# JAPAN: Joint Adaptive Prediction Areas with Normalising-Flows

**Eshant English**[1,2,3,*] **Christoph Lippert**[1,4]

[1]Hasso Plattner Institute for Digital Engineering, Germany
[2]University of Tokyo, Tokyo, Japan
[3]Centre for Advanced Intelligence Project, RIKEN, Tokyo, Japan
[4]Hasso Plattner Institute for Digital Health at Mount Sinai, USA

## Abstract

Conformal prediction provides a model-agnostic framework for uncertainty quantification with finite-sample validity guarantees, making it an attractive tool for constructing reliable prediction sets. However, existing approaches commonly rely on residual-based conformity scores, which impose geometric constraints and struggle when the underlying distribution is multimodal. In particular, they tend to produce overly conservative prediction areas centred around the mean, often failing to capture the true shape of complex predictive distributions. In this work, we introduce **JAPAN** (*Joint Adaptive Prediction Areas with Normalising-Flows*), a flow-based framework that uses density estimates for several conformal scores. By leveraging flow-based models, JAPAN estimates the (predictive) density and constructs prediction areas by thresholding on the estimated density scores, enabling compact, potentially disjoint, and context-adaptive regions that retain finite-sample coverage guarantees. We theoretically motivate the efficiency of JAPAN and empirically validate it across multivariate regression and forecasting tasks, demonstrating good calibration and tighter prediction areas compared to existing baselines. Furthermore, the several *density-based conformity scores* showcase the flexibility of our proposed framework.

## 1 Introduction

Conformal Prediction (CP) (Gammerman et al., 1998; Shafer & Vovk, 2008) offers a model-agnostic framework for uncertainty quantification with finite-sample coverage guarantees. For a user-specified significance level $\epsilon \in (0, 1)$, CP constructs a prediction region $\Gamma_\epsilon(x) \subseteq \mathcal{Y}$ (the label space for a prediction $y$) satisfying $\mathbb{P}(y \in \Gamma_\epsilon(x)) \geq 1 - \epsilon$, i.e. the true output lies in the prediction set with a high probability given by the significance level $\epsilon$. This makes CP an appealing choice for enhancing the reliability of predictive models, particularly in safety-critical or high-stakes applications.

Traditionally, CP has been implemented using residual-based conformity scores, such as the absolute error $|y - \hat{y}|$. While intuitive and effective for univariate response $y$, it does not remain so in higher dimensions as the residuals become multivariate. The residuals can be mapped to a univariate conformity score, but this requires a choice of geometry—e.g., $\ell_2$-norms construct spherical regions, and $\ell_1$-norms give rise to rectangular regions. However, imposing such geometric constraints on the prediction area may not reflect the true nature of the underlying uncertainty. Additionally, residual-based conformity scores tend to over-concentrate around a single mode, resulting in overly broad sets when the predictive distribution is multimodal.

---

[*]contact email: eshant.english@hpi.de, eshantenglish@g.ecc.u-tokyo.ac.jp

Figure 1: For the spiral density, only JAPAN produces compact regions that align with high-density areas, as the prediction area closely follows the spiral, whereas other baselines fail to capture this.

Although CP guarantees *validity*, practical usefulness also requires *efficiency*: the size of the prediction region. Ideally, we seek the *smallest possible region* that maintains coverage, tightly capturing likely outcomes while excluding unnecessary low-density areas. It is already known (Lei et al., 2013) that conformal prediction areas obtained by thresholding the conditional density $p(y \mid x)$ are compact, and can even be the smallest possible when the true threshold ($\tau_\epsilon$) is not a function of $x$. While the conditional density $p(y \mid x)$ can yield efficient conformal areas in theory, the density is typically unknown in practice. Fortunately, it can be approximated using estimators such as *flow-based models* (Papamakarios et al., 2021; Dinh et al., 2017), which are flexible generative models that learn a mapping either explicitly or implicitly from a base distribution to a target distribution. They have a tractable *likelihood* and can be used to approximate $p(y \mid x)$. Additionally, as we will see, their latent space sampling provides an efficient mechanism for computing the regions, making them attractive in comparison to other density estimation methods such as Kernel Density Estimation (KDE). When the estimated density $\hat{p}(y \mid x)$ is sufficiently accurate, the resulting conformal prediction sets approach the size under the true density threshold while preserving validity. This insight motivates a shift away from distance-based conformity scores and towards density-based ones.

In this work, we propose **JAPAN** (*Joint Adaptive Prediction Areas for Normalising-flows*), a method that uses the (conditional) density estimates as the conformity score computed by *flow-based models*. Largely, we motivate JAPAN-constructed prediction areas by thresholding on densities from a calibration set, such as: $\Gamma_\epsilon(x) := \{y \in \mathcal{Y} \ : \ \log \hat{p}(y \mid \cdot) \geq \tau_\epsilon\}$, where $\tau_\epsilon$ is the $(1 - \epsilon)$-quantile of calibration scores. These areas are **geometry-free**, avoiding assumptions like ellipsoids or rectangles; **potentially disjoint**, adapting to the shape of the predictive distribution by only aligning with areas of high density, especially when it is multimodal; and **context-adaptive**, adjusting with the input $x$, particularly in structured settings like time series.

We summarise our main contributions as follows:

- We propose **JAPAN**, a flow-based method that uses log-densities as conformity scores, enabling efficient, adaptive, and potentially multimodal prediction areas.
- We theoretically show that the regions formed are close to the ones formed by the true unconditional threshold under mild ranking errors.
- We empirically evaluate JAPAN across diverse tasks—multivariate regression and time series forecasting (while also proposing a new architecture)—demonstrating good coverage and tighter areas compared to other baselines.
- We highlight the versatility of JAPAN by several *density-based conformity scores*, including unconditional, conditional, posterior-based, latent-based, and $\tau_\epsilon$ adaptive modelling setups.

Throughout the paper, we use the term *area* as a measure of prediction sets, though in higher dimensions this naturally generalises to volume. Similarly, we use *Normalising Flows (NFs)* as an umbrella term for the broader class of *flow-based generative models*.

## 2 PRELIMINARIES

### 2.1 NORMALISING FLOWS

**Normalising Flows (NFs)** are a class of generative models that transform a simple base distribution into a complex target distribution either by a sequence of bijective mappings or by integrating continuous dynamics (e.g., via learned velocity fields or score functions). This includes discrete flows as well as continuous-time flows such as *score-based diffusion models* (bijective mapping exists through the probability flow ODE) (Song et al., 2021; 2020), *continuous normalising flows (CNFs)*

(Grathwohl et al., 2018) trained with or without *Flow-Matching* (Lipman et al., 2022), or *stochastic interpolants* objectives (Albergo & Vanden-Eijnden, 2022).

Formally, a NF can realise a bijective mapping $h : \mathbb{R}^{D_y} \to \mathbb{R}^{D_y}$, such that $z = h(y) \sim p_Z(z)$, where $p_Z$ is a simple base distribution, typically standard Gaussian. The mapping can be explicitly learned by a sequence of invertible transformations Dinh et al. (2017) or obtained by integrating an ODE defined by a learned velocity field (Lipman et al., 2022), or the score function (Song et al., 2020). For NF the density of $y$ can be computed by the change-of-variables formula:

$$\log p(y \mid x) = \log p_Z(h(y, x)) + \Phi(y, x). \tag{1}$$

where $\Phi(y, x)$ encodes the change in log-volume change due to the underlying flow mapping, e.g., a log-determinant Jacobian for discrete flows (Dinh et al., 2017), or a time-integrated divergence term for continuous-time models (Chen et al., 2018). For the unconditional case, we simply remove the context $x$ and have $h, \Phi$ only depend on $y$, and thus yield $p(y)$ with a similar log-likelihood equation.

## 2.2 INDUCTIVE CONFORMAL PREDICTION

**Inductive Conformal Prediction (ICP)** (Papadopoulos et al., 2002; Vovk et al., 2005) is a scalable variant of conformal prediction that enables valid prediction sets by splitting the data into training and calibration sets. Let $\mathcal{D} = \{(x_i, y_i)\}_{i=1}^n$ be a dataset of i.i.d. samples from an unknown distribution $\mathbb{P}$. This dataset is partitioned into a *training set* $\mathcal{D}_{\text{train}} = \{(x_i, y_i)\}_{i=1}^m$ and a *calibration set* $\mathcal{D}_{\text{cal}} = \{(x_i, y_i)\}_{i=m+1}^n$, where $n = m + l$.

Typically, a regression model $h : \mathcal{X} \to \mathcal{Y}$ is fitted on $\mathcal{D}_{\text{train}}$, and a *conformity score* $\alpha_i$ is computed on each calibration point $(x_i, y_i)$. For a test input $x^*$ and a candidate output $y$, the test conformity score $\alpha^*$ is compared against the calibration scores via a p-value:

$$\pi(y) = \frac{1}{l + 1} \left( 1 + \sum_{i=1}^l \mathbb{1}(\alpha_i \le \alpha^*) \right),$$

which quantifies how plausible $y$ is under the model and calibration set. The $1 - \epsilon$ prediction region is then defined as:

$$\Gamma_\epsilon(x^*) = \{y \in \mathcal{Y} \mid \pi(y) > \epsilon\}.$$

In the classical univariate regression setting, it is natural to work with nonconformity score, i.e. testing the implausibility of a label instead, and is defined as the absolute residual: $\alpha_i = |y_i - \mu(x_i)|$ with $\mu$ as the predictor. In this case, the prediction set $\Gamma_\epsilon(x^*)$ can be constructed directly as a symmetric interval centred at $\mu(x^*)$. This interval can be obtained via a simple inversion of the above p-value condition for the residual score such that $\Gamma_\epsilon(x^*) := \{\mu(x^*) \pm q_{1-\epsilon}\}$ ,, where $q_{1-\epsilon}$ is the $(1 - \epsilon)$-quantile of the nonconformity scores $\{\alpha_i\}_{i=1}^l$ based on the absolute residual error.

## 3 JAPAN: JOINT ADAPTIVE PREDICTION AREAS VIA NORMALISING-FLOWS

We apply **JAPAN** to two settings: multivariate regression and exchangeable trajectories of time series. Despite differing in structure, from a conformal inference perspective, both share the same core challenge: designing conformity scores that yield valid and *efficient* prediction areas. A particular difficulty in multivariate outputs, especially for time series, is the presence of inter-dimensional (temporal) dependencies. These make simple geometric constructions like L2 balls or Bonferroni-corrected rectangular areas inefficient, as they do not incorporate inter-dimensional dependencies. Since JAPAN works directly with the distribution of $\mathbb{P}(Y \mid \cdot)$ by leveraging normalising flows, we can work with both settings with minimal changes. Note that JAPAN does not require a Normalising Flow to be trained conditionally on $X$, as we will see later in Section 5, or Section A, Normalising Flows can be trained unconditionally, or conditioned on $\hat{Y}$, or like a generative model $(\cdot|Y)$, however, we use the notion $\mathbb{P}(Y \mid \cdot)$ first for simplicity. Also, we have a slight abuse notation where $x, y, z$ are variables and the subscript versions specify specific instances of the variables, i.e. $x_i, y_i, z_i$.

### 3.1 GENERAL SETTING: MULTIVARIATE REGRESSION

Let $x \in \mathbb{R}^{D_x}$ denote the covariate (conditioning) variable and $y \in \mathbb{R}^{D_y}$ the multivariate response. Our goal is to construct a prediction set $\Gamma_\epsilon(x) \subset \mathbb{R}^{D_y}$ such that

$$\mathbb{P}(y \in \Gamma_\epsilon(x)) \ge 1 - \epsilon,$$

with $\Gamma_\epsilon(x)$ being as compact as possible. JAPAN models the conditional distribution $p(y \mid x)$ using a Normalising Flow (NF), which realises the following bijective mapping:

$$z = h(y, x), \quad y = h^{-1}(z, x), \quad z \sim p_Z(z),$$

where $p_Z$ is a base distribution (typically standard Gaussian).

We compute the log-density from this approximation as the conformity score as defined in Eq. (1). After training the NF on the training set, we compute conformity scores on the held-out calibration set. For any candidate test point $(x^*, y)$, we define the conformal prediction set as:

$$\Gamma_\epsilon(x^*) = \left\{ y \in \mathbb{R}^{D_y} \ : \ \log \hat{p}(y \mid x^*) \geq \tau_\epsilon \right\},$$

where $\tau_\epsilon$ is the $(1 - \epsilon)$-quantile of the log-densities observed on the calibration set. Section 3.3 shows the full algorithm. This region satisfies the CP coverage guarantee by construction (Vovk et al., 2023) and, under assumptions such as preservation of rank (Proposition 1) and approximate rank preservation (Proposition 2), the prediction areas closely approximate the prediction areas realised by thresholding on the true unknown density. Proposition 4 shows how the inaccuracies in normalising flows estimation impact the efficiency of the computed area.

**Proposition 1** (Optimality under Rank Preservation). *Let $p(y \mid x)$ be the true conditional density, and let $f_\theta(y \mid x)$ be the model-estimated density. If there exists a strictly increasing function $g : \mathbb{R}^+ \to \mathbb{R}^+$ such that $f_\theta(y \mid x) = g(p(y \mid x))$ for all $y$, then the conformal prediction region defined by $f_\theta$ is the same as under the true density:*

$$Area(\Gamma_\epsilon(x)) = Area(\Gamma_\epsilon^*(x)).$$

**Proposition 2** (Approximate Optimality under Approximate Rank Preservation). *Assume $f_\theta(y \mid x)$ approximates a monotonic transformation of $p(y \mid x)$ within a uniform error $\delta$, i.e., $|f_\theta(y \mid x) - g(p(y \mid x))| \leq \delta$ for a strictly increasing $g$. Then the area difference between the predicted and optimal regions is bounded:*

$$|Area(\Gamma_\epsilon(x)) - Area(\Gamma_\epsilon^*(x))| \leq C(\delta),$$

*where $C(\delta) \to 0$ as $\delta \to 0$.*

### 3.2 SPECIAL CASE: MULTI-STEP TIME SERIES TRAJECTORIES

JAPAN naturally extends to uncertainty quantification for predicting future of time series trajectories by treating the conditioning variable $x$ as a temporal context window. Here, we have access to a collection of independent time-series trajectories, where each entire trajectory is treated as one data point. All trajectories are assumed to be sampled from the same underlying distribution, meaning that the resulting conformity scores are exchangeable and do not suffer from serial dependence. Note that this setting is different from the one where we have access to a single time series and each time index is a data point and exchangeability is violated. Note that this setting yields exchangeable conformity scores and by construction the prediction areas satisfies the coverage guarantees (validity) (English et al., 2024; Vovk et al., 2023).

Let $\{x_{1:T+H}^{(i)}\}_{i=1}^n$ be a collection of multivariate time series trajectories, where $x_{1:T}^{(i)} \in \mathbb{R}^{T \times D}$ is the observed history and $x_{T+1:T+H}^{(i)} \in \mathbb{R}^{H \times D}$ is the prediction target. We summarise the temporal context into a representation $c_T^{(i)}$, computed using a recurrent or attention-based encoder such as an RNN or Transformer Vaswani et al. (2017) as done in Rasul et al. (2021); Feng et al. (2023), i.e. $c_t^{(i)} = \text{RNN}(x_{t-1}^{(i)}, c_{t-1}^{(i)})$, where, $c_T^{(i)} = \text{context encoder}(x_{1:T}^{(i)})$.

The NF then models the conditional distribution:

$$z^{(i)} = h(x_{T+1:T+H}^{(i)}, c_T^{(i)}), \quad x_{T+1:T+H}^{(i)} = h^{-1}(z^{(i)}, c_T^{(i)}), \quad z^{(i)} \sim p_Z(z),$$

where $h$ is a bijective transformation realised by the flow mapping and is conditioned on the context vector $c_T^{(i)}$. The conditional density $p(x_{T+1:T+H}^{(i)} \mid x_{1:T}^{(i)})$ can be evaluated similar to $(x_i, y_i)\}_{i=1}^n$ pairs in the multivariate regression case in Eq. (1) and used as the conformity score.

### 3.3 Estimating prediction area

Formally, we want to estimate the area of the conformal set

$$\Gamma_\epsilon(x) := \left\{ y \in \mathbb{R}^{D_y} \ : \ \log \hat{p}(y \mid x) \geq \tau_\epsilon \right\},$$

where $\tau_\epsilon$ is the density threshold obtained from the calibration scores at level $1 - \epsilon$.

While evaluating coverage in JAPAN is straightforward—requiring only log-likelihood evaluation and thresholding—the computation of the prediction area is not. One straightforward way can be with methods like Monte Carlo or Quasi-Monte Carlo, but they can be computationally expensive, especially if the support of the label space $\mathcal{Y}$ is unknown. Fortunately, due to the bijective structure of normalising flows, we can shift the area estimation to the latent space, which allows for efficient computation as in Proposition 3 (see Table 10 for run time comparison against naive MC sampling).

**Proposition 3** (Area/Volume computation). *Let $z \sim p_Z$ be samples from the base distribution and let $y = h^{-1}(z, x)$ be the realised inverse flow transformation conditioned on $x$, and $\exp(\phi(z, x))$ be area/volume correction under the inverse mapping. Then the area of the prediction set $\Gamma_\epsilon(x)$ can be computed as*

$$\widehat{\mathrm{Area}}(\Gamma_\epsilon(x)) = \frac{1}{N} \sum_{i=1}^{N} \mathbb{1}\left\{ \hat{p}(h^{-1}(z_i, x) \mid x) \geq \tau_\epsilon \right\} \cdot \frac{\exp(\phi(z_i, x))}{p_Z(z_i)}.$$

This estimator accounts for the transformation from latent to data space and includes only those latent samples whose transformed outputs fall within the conformal region. Since the base distribution $p_Z$ is typically standard Gaussian, it can be sampled from efficiently, making this approach practical even in moderately high-dimensional settings. See Section B for the proofs. Additionally, the estimator is linked to importance sampling (see Section B.1) which also shows why the estimator is efficient and has low variance.

---

**Algorithm 1** JAPAN: Joint Adaptive Prediction Areas with Normalising-Flows

---

**Require:** Training set $\mathcal{D}_{\text{train}} = \{(x_i, y_i)\}_{i=1}^{n}$, calibration set $\mathcal{D}_{\text{cal}} = \{(x_j, y_j)\}_{j=1}^{l}$, significance level $\epsilon \in (0, 1)$
**Ensure:** Prediction set $\Gamma_\epsilon(x)$ satisfying marginal coverage $\geq 1 - \epsilon$
1: Train a conditional normalising flow to estimate $\hat{p}(y \mid x)$ on $\mathcal{D}_{\text{train}}$:
2: **for** each calibration example $(x_j, y_j) \in \mathcal{D}_{\text{cal}}$ **do**
3:    Compute conformity score: $\alpha_j = \log \hat{p}(y_j \mid x_j)$ based on Eq. (1)
4: **end for**
5: Compute threshold $\tau_\epsilon$ as the $\lfloor \epsilon \cdot (1 - 1/m) \cdot m \rfloor$-th smallest value in $\{\alpha_j\}_{j=1}^{m}$
6: **At test time:** For new input $x$, define the prediction are using Proposition 3 such that:

$$\Gamma_\epsilon(x) = \{y \in \mathcal{Y} \ : \ \log \hat{p}(y \mid x) \geq \tau_\epsilon\}$$

---

## 4 Experiments

We evaluate **JAPAN** on various datasets (multivariate and time series) and against various baselines with normalising flows used as a base predictor. In particular, we use CONTRA (Fang et al., 2024), PCP (Wang et al., 2023), RCP (Johnstone & Cox, 2021), NLE (Messoudi et al., 2020), MCQR (Fang et al., 2024), DIST-SPLIT (Izbicki et al., 2020), CQR (Romano et al., 2019), COPULACPTS (Sun & Yu, 2024), VANILACOPULA (Messoudi et al., 2021), CRNN (Stankeviciute et al., 2021), JANET (English et al., 2024) as baselines. All the baselines used either official implementation or public implementations closely following the set up from the original works. See Table 1 for a qualitative comparison of all the baselines.

In our experiments, we exclusively use *discrete normalising flows* (Dinh et al., 2017)—that is, flows composed of a finite sequence of explicit invertible transformations such as affine (Kingma & Dhariwal, 2018; English et al., 2023), spline (Durkan et al., 2019), or autoregressive Papamakarios et al. (2018). These models allow for efficient sampling and exact log-likelihood evaluation without requiring the numerical integration of ODEs. Moreover, recent works such as TARFLOW (Zhai

| Property | JAPAN | CONTRA | PCP | JANET | Dist-Split | NLE | CQR | MCQR | RCP | CopulaCPTS | VanillaCopula | CFRNN |
|---|---|---|---|---|---|---|---|---|---|---|---|---|
| **Disjoint** | ✓ | ✗ | ✓ | ✗ | ✗ | ✗ | ✗ | ✗ | ✗ | ✗ | ✗ | ✗ |
| **Geometry free** | ✓ | △ | △ | ✗ | ✗ | ✗ | ✗ | ✗ | ✗ | ✗ | ✗ | ✗ |
| **Context-based** | ✓ | ✓ | ✓ | ✓ | ✓ | ✓ | ✗ | ✗ | ✗ | ✗ | ✗ | ✗ |

Table 1: Comparisons of different baselines, where ✓ = Yes; △ = partially/soft geometric constraint; ✗ = No. JAPAN is the only method that can provide disjoint regions, is context-adaptive, and has no constraint on the geometry, whereas PCP and CONTRA have soft constraints on the geometry.

et al., 2024) demonstrated that with sufficiently expressive architectures, their generation quality and likelihood estimation rival those of continuous-time flow models and score-based diffusion models. For discrete models, the conditional density is computed via the change-of-variables formula (Winkler et al., 2019):

$$\log p(y \mid x) = \log p_Z(h(y, x)) + \log \left| \det \left( \frac{\partial h(y, x)}{\partial y} \right) \right|.$$

**Time series modelling:** While any normalising flow architecture can be used in principle, during modelling it is important to incorporate the *causal structure* inherent to time series data. In practice, architectures that preserve the temporal relationship tend to perform better. To this end, we adapt the TARFLOW architecture (Zhai et al., 2024), originally for image data, for time series modelling. Further architectural details are provided in Section C.1.

## 4.1 TOY DENSITIES

To illustrate the behaviour of different conformal prediction methods, we visualise prediction areas on a few synthetic 2D distributions. These datasets have sharp boundaries and disconnected high-density regions, serving as a test-bed for evaluating how well different methods adapt to complex geometries and avoid low-density areas.

Figure 1 shows a visualisation of different regions on a *spiral* density, **JAPAN** constructs the most compact prediction area, closely following the spiral density and effectively avoiding low-density regions. This reflects its ability to threshold directly on the estimated density in the data space, without imposing geometric constraints. **CONTRA**, while geometry-free in data space, constructs spherical regions in the latent space. As a result, low-density regions in the data space that correspond to high-density latent samples are mistakenly included in the conformal region. **PCP** also allows for disjoint regions and avoids global geometric constraints, but its local ball-based area construction introduces soft constraints that fail on densities with ring-like modes. In this case, co-eccentric balls cover both dense and non-dense regions indiscriminately. **Elliptical** and **Rectangular** conformal methods impose hard geometric assumptions, leading to highly inefficient prediction areas that encapsulate large regions of near-zero density. Computed area along with visualisation of other toy densities can be found in Section A.1

## 4.2 MULTIVARIATE REGRESSION

We evaluate on four multivariate regression benchmarks: **Energy** (Tsanas & Xifara, 2012), **RF2D** (Spyromitros-Xioufis et al., 2016), **RF4D** (Spyromitros-Xioufis et al., 2016), and **SCM20D** (Spyromitros-Xioufis et al., 2016). These tasks span domains from energy efficiency modelling, river flow forecasting, to structured output prediction under supply chain simulation. We repeated the experiments over *25 random data splits*. Details on the datasets and experiments can be found in Section D and Section C.2

As shown in Table 2, most methods—including CQR, MCQR, RCP, DIST-SPLIT, NLE, CFRNN, and VANILLACOPULA—achieve comparable marginal coverage across datasets. Figure 2 shows area plots for five methods, with the smallest areas with **JAPAN** consistently yielding the smallest prediction areas, demonstrating strong volume efficiency without compromising coverage. This advantage is particularly pronounced in a relatively high-dimensional setting of *RF4D*. We also note that **JANET** and **CopulaCPTS** perform poorly in some settings, leading to high standard deviations. This is due to these methods using half of the calibration set to train an auxiliary copula or residual predictor model, and their final prediction intervals can become numerically unstable. In particular, when the secondary model produces near-zero outputs used as denominators for score rescaling, the resulting prediction regions may explode, leading to highly inflated areas in some runs.

| Method | Cov. (Energy) | Area | Cov. (RF2D) | Area | Cov. (RF4D) | Area | Cov. (SCM) | Area (×10³) |
|---|---|---|---|---|---|---|---|---|
| CONTRA | 0.88 ± 0.03 | 18.81 ± 7.88 | 0.91 ± 0.01 | 5.33 ± 0.52 | 0.89 ± 0.01 | 59.27 ± 18.60 | 0.89 ± 0.01 | 61.93 ± 5.70 |
| PCP | 0.88 ± 0.03 | 16.58 ± 3.87 | 0.91 ± 0.01 | 7.39 ± 1.02 | 0.91 ± 0.01 | 111.63 ± 18.93 | 0.89 ± 0.01 | 68.75 ± 7.75 |
| NLE | 0.87 ± 0.03 | 21.90 ± 9.06 | 0.91 ± 0.01 | 15.47 ± 1.28 | 0.90 ± 0.01 | 2732.15 ± 1598.94 | 0.90 ± 0.01 | 102.13 ± 21.19 |
| RCP | 0.86 ± 0.04 | 30.79 ± 17.68 | 0.91 ± 0.01 | 23.86 ± 3.24 | 0.90 ± 0.01 | 4939.29 ± 4175.90 | 0.89 ± 0.01 | 64.69 ± 5.61 |
| MCQR | 0.88 ± 0.03 | 28.99 ± 11.83 | 0.91 ± 0.01 | 12.07 ± 0.97 | 0.91 ± 0.01 | 1034.89 ± 2279.10 | 0.91 ± 0.01 | 70.93 ± 6.85 |
| Dist-Split | 0.86 ± 0.03 | 35.36 ± 12.64 | 0.91 ± 0.01 | 19.22 ± 1.60 | 0.91 ± 0.01 | 1595.92 ± 6355.70 | 0.91 ± 0.01 | 87.10 ± 9.23 |
| CQR | 0.88 ± 0.02 | 31.12 ± 12.05 | 0.91 ± 0.01 | 12.50 ± 1.01 | 0.91 ± 0.01 | 1180.65 ± 9536.00 | 0.91 ± 0.01 | 84.48 ± 8.74 |
| CopulaCPTS | 0.90 ± 0.03 | 67.20 ± 49.72 | 0.91 ± 0.01 | 25.98 ± 4.19 | 0.91 ± 0.01 | 1591.26 ± 6631.00 | 0.89 ± 0.02 | 71.57 ± 9.09 |
| VanilaCopula | 0.86 ± 0.03 | 39.50 ± 30.91 | 0.91 ± 0.01 | 25.68 ± 3.73 | 0.89 ± 0.01 | 1363.76 ± 5296.00 | 0.91 ± 0.01 | 67.84 ± 6.38 |
| CFRNN | **0.90 ± 0.03** | **56.22 ± 44.29** | 0.91 ± 0.01 | 27.15 ± 3.71 | **0.92 ± 0.01** | 3322.47 ± 14397.00 | 0.91 ± 0.01 | 83.60 ± 7.81 |
| JANET | 0.88 ± 0.07 | 42.69 ± 45.52 | 0.88 ± 0.02 | 21.87 ± 13.20 | 0.85 ± 0.10 | 1002.33 ± 3237.00 | 0.90 ± 0.01 | 64.30 ± 21.06 |
| JAPAN | 0.88 ± 0.02 | 16.32 ± 1.39 | **0.91 ± 0.01** | **5.06 ± 0.41** | **0.91 ± 0.01** | **24.11 ± 2.33** | **0.90 ± 0.01** | **61.28 ± 5.72** |

Table 2: Coverage and prediction areas for the **Energy**, **RF2D**, **RF4D**, and **SCM** datasets computed over *25 random data splits*. Areas for SCM are shown in units of $\times 10^3$. Almost all methods achieve the desired coverage level of 0.9. We bold the methods for each datasets that have a mean coverage of 0.90±0.01 and has the lowest area for the dataset. Thus, JAPAN is bolded in all datasets except **Drone**, but has a smaller volume of all with only a 0.02 difference with the bolded **CFRNN** method.

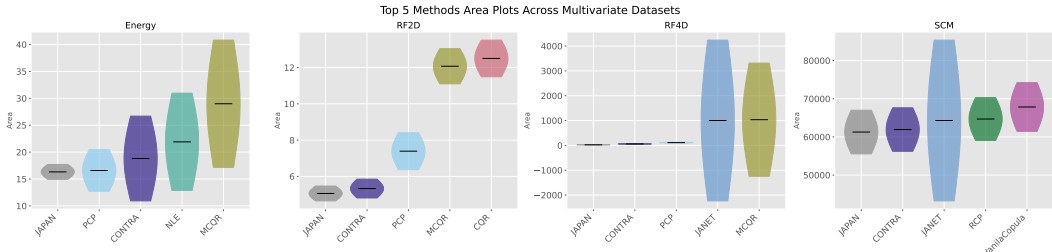

Figure 2: Prediction areas per dataset (Energy, RF2D, RF4D, SCM) for the top 5 methods computed over *25 random data splits*. Each bar's length signifies the standard deviations, with methods sorted by average area for each dataset. JAPAN consistently has the lowest areas and hence is more informative.

## 4.3 TIME SERIES DATA

We also evaluated on four time series datasets: **COVID-19** (Stankeviciute et al., 2021), **Particle-1** (Kipf et al., 2018), **Drone** (Sakai et al., 2018), and **Pedestrian** (van den Berg et al., 2008), each representing different forecasting challenges in scientific or physical systems. These datasets include multi-step prediction tasks where temporal context plays a central role. We repeated the experiment over *25 random data splits*. Further details on the dataset and experiments, such as input windows, horizons, and preprocessing, are provided in Section D and Section C.3.

As shown in Table 3, all methods achieve similar marginal coverage across datasets. Figure 3 shows area plots for five methods, with the smallest areas with **JAPAN** consistently producing the most compact prediction regions, outperforming all the baselines. Notably, JAPAN yields up to an order-of-magnitude smaller volumes in COVID-19 while maintaining reliable coverage. On the other hand, methods such as CQR, MCQR, and CFRNN tend to generate significantly larger regions. We also observe **RCP** produce unreliable results in some cases. RCP suffers from numerical instability and undefined scores in some settings, as reflected by missing volume (due to infinity) for COVID-19 data. Extended Plots for Figure 2 and Figure 3, and the calibration curve aggregated over all datasets for all methods can be found in Section A.

## 5 EXTENSIONS: APPLICATIONS OF OTHER DENSITY-BASED SCORES

Now, we present several extensions that demonstrate the flexibility of using normalising flows for constructing prediction areas. In our main experiments, we used **JAPAN** as both a predictor and an estimator of log-densities. However, there are practical situations where one may already have a trained predictive model or may wish to use a different base model due to task-specific reasons. In such cases, JAPAN can still be applied on top of the base model and thus decouple the predictive mechanism from the density estimation. This can be viewed as using a *fixed context encoder*, whose representation is learned through an external or unrelated training objective, or could be trained under a single objective.

| Method | Cov. (COVID-19) | Area | Cov. (Particle-1) | Area | Cov. (Drone) | Area | Cov. (Pedestrian) | Area |
|---|---|---|---|---|---|---|---|---|
| CONTRA | 0.92 ± 0.03 | 563.64 ± 98.31 | 0.87 ± 0.01 | 0.90 ± 0.01 | **0.89 ± 0.02** | **1.51 ± 0.12** | 0.89 ± 0.02 | 0.55 ± 0.03 |
| PCP | 0.91 ± 0.03 | 610.56 ± 359.73 | 0.87 ± 0.01 | 1.71 ± 0.04 | 0.89 ± 0.03 | 3.21 ± 0.04 | 0.88 ± 0.02 | 1.60 ± 0.03 |
| NLE | 0.90 ± 0.04 | 416.13 ± 57.55 | 0.87 ± 0.01 | 2.31 ± 0.00 | 0.88 ± 0.02 | 3.12 ± 0.08 | 0.88 ± 0.03 | 2.23 ± 0.02 |
| RCP | 0.87 ± 0.03 | – | 0.88 ± 0.02 | 2.28 ± 0.01 | 0.88 ± 0.03 | 3.16 ± 0.06 | 0.87 ± 0.02 | 2.23 ± 0.03 |
| MCQR | 0.91 ± 0.03 | 1276.50 ± 222.80 | 0.91 ± 0.02 | 2.89 ± 0.04 | 0.86 ± 0.03 | 3.32 ± 0.12 | 0.86 ± 0.02 | 1.89 ± 0.05 |
| Dist-Split | 0.85 ± 0.03 | 652.60 ± 80.67 | 0.96 ± 0.01 | 3.28 ± 0.07 | 0.93 ± 0.01 | 4.47 ± 0.28 | 0.91 ± 0.03 | 2.20 ± 0.07 |
| CQR | 0.93 ± 0.02 | 1048.69 ± 175.68 | 0.96 ± 0.01 | 3.33 ± 0.06 | 0.98 ± 0.03 | 5.12 ± 0.74 | 0.96 ± 0.02 | 2.29 ± 0.15 |
| CopulaCPTS | 0.85 ± 0.07 | 627.72 ± 167.06 | 0.89 ± 0.03 | 3.15 ± 0.12 | 0.92 ± 0.05 | 4.46 ± 0.73 | 0.91 ± 0.03 | 2.19 ± 0.26 |
| VanilaCopula | 0.84 ± 0.04 | 539.09 ± 79.93 | 0.87 ± 0.02 | 2.91 ± 0.05 | 0.86 ± 0.02 | 3.48 ± 0.08 | 0.87 ± 0.02 | 1.91 ± 0.05 |
| CFRNN | 0.92 ± 0.03 | 927.09 ± 112.44 | 0.97 ± 0.01 | 3.39 ± 0.09 | 0.99 ± 0.01 | 5.53 ± 0.26 | 0.97 ± 0.02 | 2.47 ± 0.24 |
| JANET | 0.88 ± 0.04 | 669.10 ± 364.47 | 0.92 ± 0.02 | 2.78 ± 0.09 | 0.90 ± 0.06 | 3.68 ± 0.26 | 0.92 ± 0.03 | 2.20 ± 0.23 |
| JAPAN | **0.91 ± 0.03** | **400.94 ± 74.39** | **0.91 ± 0.03** | **0.89 ± 0.02** | 0.88 ± 0.02 | 1.47 ± 0.11 | **0.89 ± 0.02** | **0.50 ± 0.04** |

Table 3: Coverage and prediction areas on the **COVID-19**, **Particle-1**, **Drone**, and **Pedestrian** time series datasets computed over *25 random data splits*. Almost all methods achieve the desired coverage level of 0.9. We bold the methods for each datasets that have a mean area of 0.90±0.01 and has the lowest area for the dataset. Thus, JAPAN is bolded in all datasets except **Drone**, but has a smaller volume of all with only a 0.01 difference with the bolded **CONTRA** method.

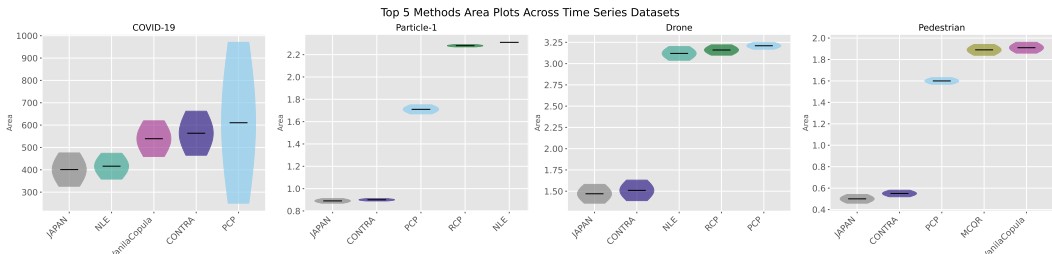

Figure 3: Prediction areas per dataset (COVID, Particle-1, Drone, Pedestrian) for the top 5 methods computed over *25 random data splits*. Each bar's length signifies the standard deviations, with methods sorted by average area for each dataset. JAPAN consistently has the lowest areas and hence is more informative.

We explore several such applications, namely **unconditional modelling** (Section A.2), where the flow is trained independently of input features or the context to learn $p(\hat{y})$; **conditional modelling** (Section A.3), where the flow is conditioned on a representation from a pretrained base model to learn $p(y|\hat{y})$; **posterior modelling** (Section A.4), where the flow is trained to learn the posterior predictive distribution ($p(\hat{y}|y)$ or more generally, $p(x|y)$), instead of $p(x|y)$); **latent modelling** (Section A.5), where we use density of the transformed sample in the latent space, thus making CONTRA (Fang et al., 2024) as a special case of JAPAN; **adaptive $\tau_\epsilon$ modelling** (Section A.6), where we discuss adapting $\tau_\epsilon$ based on the condition and effectively getting closer to conditional coverage under mild assumptions. Here, we briefly introduce conditional modelling and posterior modelling and defer to Section A for a detailed discussion on all the extensions, and to Section A.7 for a discussion on the assumption violations with underfitted models and the corresponding ablation experiments.

## 5.1 CONDITIONAL MODELLING

In this setting, we assume access to a base predictive model $\mu$ that outputs point predictions $\hat{y} = \mu(x)$. Then, we train a conditional normalising flow to model $\mathbb{P}(Y \mid \hat{Y})$. This can be seen as obtaining a fixed representation extracted from a fixed base model and training a flow separately from the base predictor, in contrast to standard end-to-end JAPAN.

This setting creates a connection between the predictions of the base model $\hat{y}$ and the true targets $y$, allowing the resulting density to capture input-dependent uncertainty. While the addition of another model introduces potential estimation error, we find that the regions constructed via log-likelihood thresholding remain both compact and valid as the flow model tries to correct for the base predictor's distribution estimation error. Consequently, this form of conditional modelling achieves more compact regions than those obtained from other baselines. Please refer to Section A.3 for experimental analysis.

## 5.2 POSTERIOR MODELLING

In this setup, we aim to model the *posterior* distribution over input conditions given an output, i.e., $\mathbb{P}(X \mid Y)$ or as a special case $\mathbb{P}(\hat{Y} \mid Y)$, using a normalising flow. We then use the resulting log-posterior density as a conformity score. While the general framework remains similar, this formulation allows us to test different possible input conditions for a given output $Y$, and select those that are most plausible under the learned posterior.

Posterior modelling has several practical advantages. For instance, in settings where the response variable $Y$ is categorical or discrete, it is often difficult or ill-posed to model $\mathbb{P}(Y \mid X)$ directly using normalising flows. Modelling $\mathbb{P}(X \mid Y)$, however, is tractable and aligns with standard generative modelling pipelines—for example, generating images conditioned on class labels. In such cases, one can perform conformal-based calibration on posterior log-densities to produce valid prediction sets. This approach can be used both to assess how plausible a generated output is and to identify whether an observed outcome falls within a calibrated prediction set.

While it is known that using the conditional density $p(y \mid x)$ with the threshold $\tau_\epsilon$ adaptive on the condition $x$ can yield the smallest conformal region under ideal conditions, it is less clear how tight the regions are when the conformity score is based on the posterior density $p(x \mid y)$. Nevertheless, we find that this approach still produces compact and valid regions in practice, albeit typically less efficient than those obtained via conditional likelihood modelling. Please refer to Section A.4 for experimental analysis.

## 6 RELATED WORK

The literature on conformal prediction (CP) (Gammerman et al., 1998; Vovk et al., 2005; Angelopoulos & Bates, 2022) is extensive, with works on new (non-)conformity scores (Sun & Yu, 2024; English et al., 2024), going beyond exchangeability (Chernozhukov et al., 2018; Barber et al., 2023) etc. In the context of non-univariate response variables, CP research primarily branches into three directions: (i) multivariate regression with i.i.d. data, (ii) forecasting in a single long time series with long-horizon or multivariate time steps (where temporal dependence violates exchangeability of conformity scores) (Gibbs & Candès, 2021; Chernozhukov et al., 2018; Lee et al., 2025), and (iii) settings where exchangeable set of time series trajectories are available—allowing CP to be trivially applied.

In the **multivariate regression** setting, RCP (Johnstone & Cox, 2021) adapts to multivariate dependencies using covariance structure to form elliptical areas. Similar to RCP, NLE (Messoudi et al., 2020) creates elliptical areas but is context adaptive. PCP (Wang et al., 2023) leverages generative models to sample new conditional outputs and then defines prediction sets via unions of local L2 balls, with (Plassier et al., 2024) extending to get approximate conditional validity. While these avoid global shape constraints, it still relies on local spherical balls based on residual distances, which we refer to as a *soft* geometric constraint. CONTRA (Fang et al., 2024), more similar to our work, employs normalising flows but constructs conformal balls in the latent space that we again term as a *soft* constraint. This allows some flexibility but risks including low-density areas in the output space. Another work (Izbicki et al., 2021) provides a disjoint region but requires a lot of hyperparameters and can have stability issues. Similarly, ST-DQR (Feldman et al., 2023) uses conditional variational autoencoders to learn low-dimensional latent representations but does not work directly with densities. Other related methods originally for univariate response but adapted for multi-variate data by Fang et al. (2024) include DIST-SPLIT (Izbicki et al., 2020) (which estimates CDFs to construct areas), LOCALLY ADAPTIVE SPLIT CP (which introduces local calibration via splitting), and quantile-based methods like CQR (Romano et al., 2019) and MCQR (Fang et al., 2024), which are well-calibrated but limited in their ability to handle multimodality. Another work, Colombo (2024) also uses Normalising Flows in the context of conformal prediction, but they used it to train the calibration process for univariate response, i.e. on the distribution of the residuals, whereas we use only a flow model to construct a region.

**For time series trajectories**, many conformal approaches focus on structured prediction over multiple horizons. BJ-RNN (Alaa & van der Schaar, 2020) applies the jackknife+ method Barber et al. (2020) to recurrent networks, but its theoretical coverage guarantee is $1 - 2\epsilon$, which is looser than the standard $1 - \epsilon$. CF-RNN (Stankeviciute et al., 2021) assumes conditionally i.i.d. residuals and

applies Bonferroni correction (Dunn, 1961), which can become overly conservative as horizon length increases. (Cleaveland et al., 2024) proposed an LP-based approach for multi-step error modelling, but it requires an additional dataset and extensive parameter tuning. COPULACPTS (Sun & Yu, 2024) builds on Messoudi et al. (2021) to use copulas for temporal dependency modelling in multivariate time series. However, it requires two calibration sets and gradient-based training, introducing inefficiency. Recent methods like CAFHT (Zhou et al., 2024) and JANET (English et al., 2024) construct rectangular areas that are adaptive to the input context but cannot produce disjoint or multimodal areas. By contrast, JAPAN's use of normalising flows enables areas that are both context-adaptive and geometrically unconstrained, yielding compact, expressive prediction areas across both multivariate and time series settings.

A recent work (Lee et al., 2025) uses flow-matching but works in the setting where only a single time series is provided and tackles the problem of exchangeability violation, whereas JAPAN tackles a completely different problem with a different approach. CCN-JPR (English & Lippert, 2024) introduced discrete normalising flows for time-series with empirical coverage guarantees, but without area computations. JAPAN extends CCN-JPR by (i) providing a comprehensive analysis across multivariate regression and time-series settings, (ii) establishing new theoretical results, (iii) proposing novel conformity scores, and (iv) developing an efficient method for area computation.

## 7 DISCUSSION AND LIMITATIONS

Our experiments demonstrate that JAPAN is a flexible and effective framework for constructing prediction regions using normalising flows. It can be applied in a wide range of settings, including multivariate regression and time series forecasting, and can be seamlessly integrated with existing predictive models. Across tasks, we observe that JAPAN consistently achieves valid coverage while producing significantly smaller prediction regions than baseline methods, particularly in high-dimensional settings.

Despite these promising results, several limitations remain. In particular, while we discuss conditional coverage when using adaptive $\tau_\epsilon$ extension (see Section A.6), JAPAN—like all methods based on marginal conformal prediction—does not guarantee *conditional coverage*. That is, while the prediction set satisfies $\mathbb{P}(y \in \Gamma_\epsilon(x)) \geq 1 - \epsilon$ in expectation over the distribution of $x$, the coverage may vary significantly between different subpopulations or contexts. Finally, while our approach can be extended to use predictions from another model, posterior densities, or an adaptive threshold, each step introduces additional assumptions and potential estimation errors. Future work may investigate the theoretical benefits these alternatives offer.

**Reproducibility Statement:** All experimental results reported in this paper can be reproduced using the implementation details provided in Section C. Dataset descriptions and preprocessing steps are detailed in Section D. Complete proofs of all theorems and propositions are included in Section B.

**Ethics Statement:** This work does not raise any specific ethical concerns. All datasets used are publicly available, and no human or animal subjects were involved. We have adhered to the ICLR Code of Ethics throughout the preparation of this paper.

## ACKNOWLEDGMENTS

We extend our gratitude to Prof Vladimir Vovk for helpful discussions and to Thomas Gaertner, Juliana Schneider, Noel Danz, and Wei-Cheng Lai for their feedback on an earlier draft. This research was supported by the HPI Research School for Digital Health. We gratefully acknowledge funding by the Deutsche Forschungsgemeinschaft (DFG, German Research Foundation) – project number 459422098.

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

## A EXTENSIONS AND ADDITIONAL EXPERIMENTS

### A.1 TOY DENSITIES

To illustrate the behaviour of different conformal prediction methods, we evaluate them on a set of synthetic 2D distributions: **Spiral**, **Checkerboard**, **Circles**, and **Moons**. These toy densities are multimodal and feature sharp discontinuities or disconnected high-density regions, making them a strong testbed for evaluating geometric flexibility of conformal methods.

Figure 4 and Figure 1 show visualisations of the conformal prediction regions produced by different methods on these datasets. **JAPAN** consistently produces compact regions that accurately track the support of the underlying distribution, avoiding low-density areas and capturing disconnected modes. By contrast, hard-constrained methods like **Elliptical** and **Rectangular** conformal prediction yield inefficient regions that fail to capture the data geometry. **CONTRA**, while geometry-free in the data space, forms spherical regions in the latent space; these often map to big continuous shapes in the data space that may include large, low-density regions. **PCP** introduces soft geometric constraints by creating unions of balls around generated samples. While this setup allows for disconnected regions, the local ball structure can still cover vacuous space when the density is highly non-uniform..

Quantitative metrics including coverage and area at $\epsilon = 0.1$ are reported in Table 4, and area comparison is plotted in Figure 5. JAPAN consistently achieves the lowest areas while maintaining valid coverage. Additionally, calibration curves across a range of $\epsilon \in [0.05, 0.95]$ are shown in Figures 10 and 11, confirming that JAPAN remains well-calibrated across different significance levels.

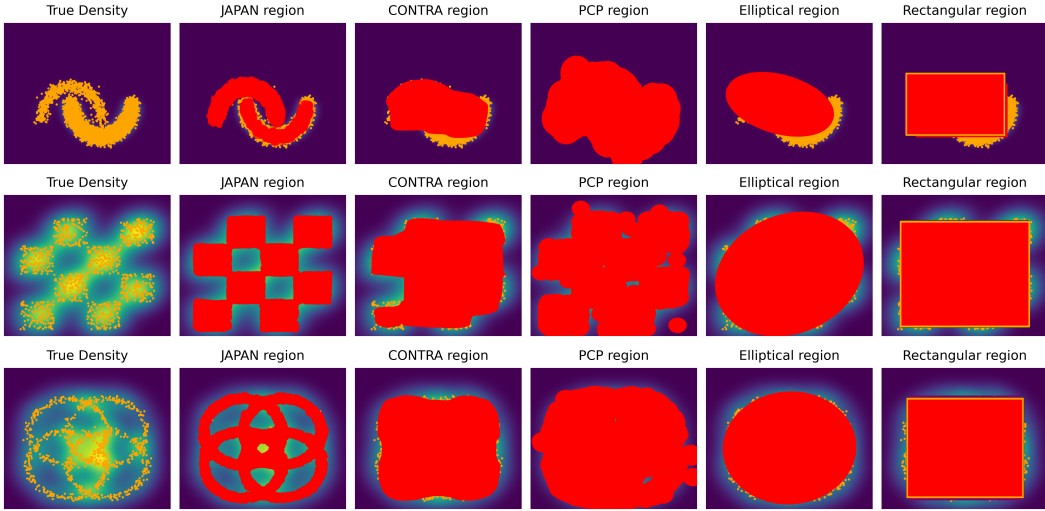

Figure 4: Visualisation of different densities (Moons, Rings, and Checkerboard) at $\epsilon = 0.1$. Only JAPAN captures the modes and avoids low-density regions. Hard constrained areas, such as Elliptical and Rectangular, naturally fail at capturing the true shape. CONTRA encapsulates low-density regions, completely ignoring discontinuities in the density. PCP while captures discontinuity includes some low-density regions due to the soft constraints.

### A.2 UNCONDITIONAL MODELLING

In the unconditional modelling setting, we assume access to a base predictive model $\mu$ that produces point predictions $\hat{y} = \mu(x)$. Rather than modelling the conditional density $p(y \mid x)$ directly, we instead fit a normalising flow to estimate the marginal distribution $\mathbb{P}(\hat{Y})$ over predicted outputs on the training set. Once the flow is trained, we apply JAPAN with a slight modification: for each calibration example $(x, y)$, we evaluate the log-density $\log \hat{p}_{\hat{Y}}(y)$ under the learned marginal model (rather than conditioning on $x$) and use this as the conformity score (see Algorithm 2). This change ensures that calibration is still performed with respect to the true output $y$.

| Method | Cov. (Checkerboard) | Area | Cov. (Moons) | Area | Cov. (Circles) | Area | Cov. (Spiral) | Area |
|---|---|---|---|---|---|---|---|---|
| CONTRA | $0.91 \pm 0.02$ | $44.04 \pm 0.45$ | $0.91 \pm 0.01$ | $3.54 \pm 0.10$ | $0.91 \pm 0.02$ | $34.06 \pm 0.10$ | $0.91 \pm 0.01$ | $25.21 \pm 0.01$ |
| PCP | $0.89 \pm 0.01$ | $32.62 \pm 0.33$ | $0.89 \pm 0.01$ | $2.47 \pm 0.09$ | $0.90 \pm 0.01$ | $18.43 \pm 0.09$ | $0.89 \pm 0.01$ | $14.28 \pm 0.01$ |
| NLE | $0.91 \pm 0.01$ | $58.24 \pm 0.42$ | $0.88 \pm 0.02$ | $4.83 \pm 0.11$ | $0.90 \pm 0.01$ | $35.93 \pm 0.11$ | $0.90 \pm 0.01$ | $26.62 \pm 2.11$ |
| RCP | $0.91 \pm 0.01$ | $57.80 \pm 0.61$ | $0.89 \pm 0.01$ | $4.84 \pm 0.13$ | $0.90 \pm 0.02$ | $35.73 \pm 0.39$ | $0.90 \pm 0.01$ | $26.38 \pm 2.64$ |
| MCQR | $0.90 \pm 0.01$ | $57.48 \pm 0.66$ | $0.89 \pm 0.01$ | $4.52 \pm 0.14$ | $0.90 \pm 0.01$ | $35.53 \pm 0.89$ | $0.90 \pm 0.02$ | $31.51 \pm 3.15$ |
| Dist-Split | $0.91 \pm 0.01$ | $57.67 \pm 0.45$ | $0.89 \pm 0.01$ | $4.70 \pm 0.09$ | $0.90 \pm 0.01$ | $35.49 \pm 0.41$ | $0.90 \pm 0.01$ | $32.64 \pm 3.26$ |
| CQR | $0.91 \pm 0.01$ | $57.77 \pm 0.21$ | $0.90 \pm 0.01$ | $4.58 \pm 0.13$ | $0.90 \pm 0.02$ | $35.56 \pm 0.36$ | $0.91 \pm 0.01$ | $32.80 \pm 3.28$ |
| JANET | $0.90 \pm 0.01$ | $57.59 \pm 0.31$ | $0.89 \pm 0.02$ | $4.57 \pm 0.12$ | $0.90 \pm 0.01$ | $35.57 \pm 0.93$ | $0.90 \pm 0.01$ | $31.57 \pm 3.16$ |
| VanilaCopula | $0.90 \pm 0.01$ | $57.17 \pm 0.36$ | $0.89 \pm 0.01$ | $4.37 \pm 0.09$ | $0.90 \pm 0.01$ | $35.39 \pm 0.34$ | $0.90 \pm 0.02$ | $32.11 \pm 2.96$ |
| CFRNN | $0.91 \pm 0.01$ | $59.11 \pm 0.52$ | $0.92 \pm 0.02$ | $4.92 \pm 0.19$ | $0.92 \pm 0.01$ | $37.49 \pm 0.71$ | $0.90 \pm 0.01$ | $34.24 \pm 3.22$ |
| **JAPAN** | $\mathbf{0.90 \pm 0.01}$ | $\mathbf{27.95 \pm 0.04}$ | $\mathbf{0.90 \pm 0.01}$ | $\mathbf{2.06 \pm 0.13}$ | $\mathbf{0.91 \pm 0.01}$ | $\mathbf{14.76 \pm 0.67}$ | $\mathbf{0.90 \pm 0.01}$ | $\mathbf{11.65 \pm 1.17}$ |
| CopulaCPTS | $0.91 \pm 0.02$ | $57.74 \pm 0.02$ | $0.89 \pm 0.01$ | $5.26 \pm 0.64$ | $0.92 \pm 0.01$ | $35.74 \pm 1.10$ | $0.90 \pm 0.01$ | $32.89 \pm 3.29$ |

Table 4: Coverage and prediction areas for the **Checkerboard**, **Moons**, **Circles**, and **Spiral** datasets computed over *25 random data splits*. Almost all methods achieve the desired coverage level of 0.9. We bold the methods for each datasets that have a mean coverage of 0.90±0.01 and has the lowest area for the dataset. Thus, JAPAN is bolded in all datasets.

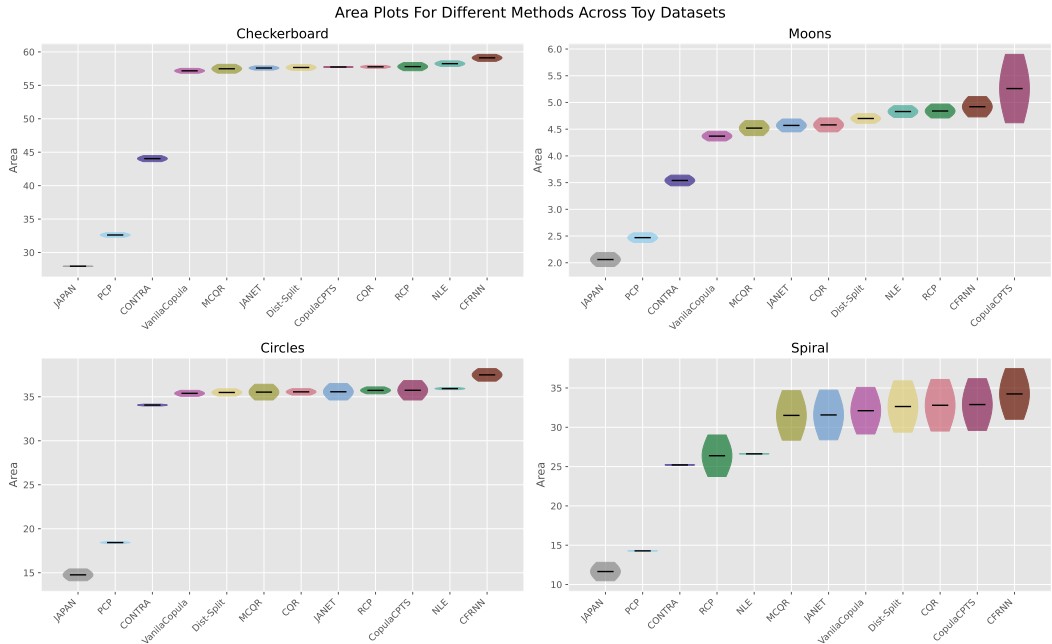

Figure 5: Computed areas per dataset (Checkerboard, Moons, Circles, Spiral) for all the methods computed over *25 random data splits*. Each bar's length signifies the standard deviations, with methods sorted by average area for each dataset. JAPAN consistently has the lowest areas and hence is more informative.

A key limitation of this approach is that the resulting prediction areas are not adaptive to the input context. Since the flow models a global distribution over outputs, all test-time regions are drawn from the same marginal. Nonetheless, we find that the method can produce surprisingly compact and valid prediction regions, especially when compared to baselines that rely on fixed geometric constraints.

We evaluate this setup on two tabular datasets: **Energy** and **RF4D**. Results are shown in Table 5. On the Energy dataset, JAPAN achieves the most compact prediction areas among all methods that satisfy the target coverage range of $0.90 \pm 0.01$. On RF4D, area efficiency is less consistent: although JAPAN remains competitive, the average area is skewed upward due to a single run with unusually large volume. This effect is even more pronounced in baselines like CONTRA and PCP, which also use latent-space conformity but without density thresholding. In contrast to JAPAN, these methods produce highly inflated regions due to poorly calibrated or geometrically mismatched latent representations. Meanwhile, standard baselines such as CQR, MCQR, also performed worse than JAPAN, despite not having the additional flow modelling. Table 5 summarises the coverage and

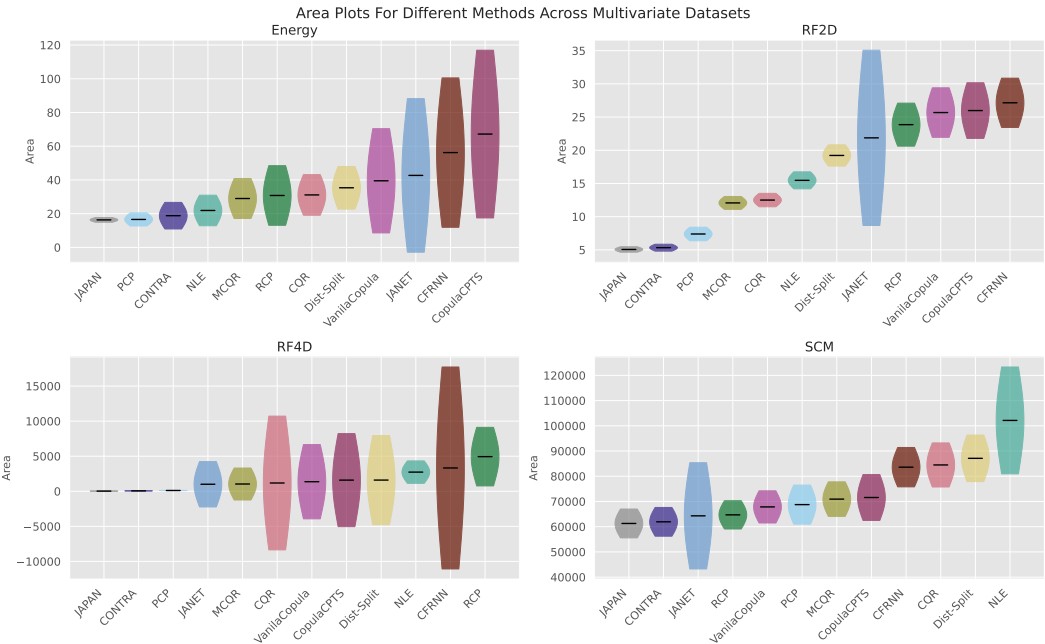

Figure 6: Prediction areas per dataset (Energy, RF2D, RF4D, SCM) for all the methods computed over *25 random data splits*. Each bar's length signifies the standard deviations, with methods sorted by average area for each dataset. JAPAN consistently has the lowest areas and hence is more informative.

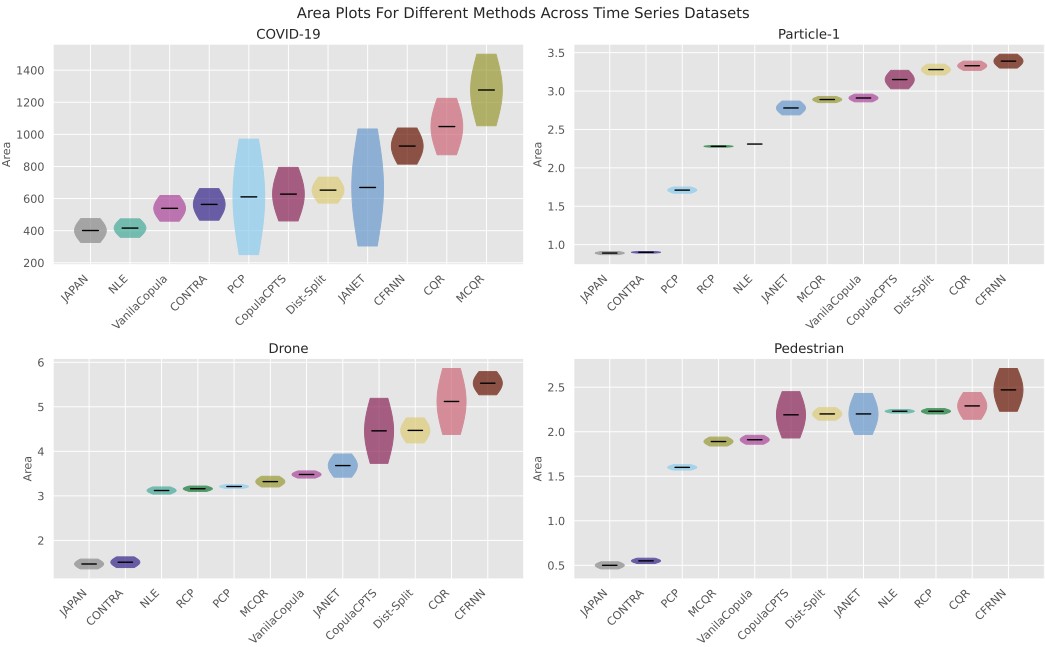

Figure 7: Prediction areas per dataset (COVID, Particle-1, Drone, Pedestrian) for all the methods computed over *25 random data splits*. Each bar's length signifies the standard deviations, with methods sorted by average area for each dataset. JAPAN consistently has the lowest areas and hence is more informative.

area metrics computed over 5 random splits. Methods with the most compact regions among those achieving near-nominal coverage are highlighted.

| Method | Cov. (Energy) | Area (Energy) | Cov. (RF4D) | Area (RF4D) x $10^4$ |
|---|---|---|---|---|
| CONTRA | 0.90 ± 0.02 | 2794.48 ± 2420.62 | 0.90 ± 0.01 | 933333.20 ± 1229039.0 |
| PCP | 0.89 ± 0.01 | 1612.79 ± 2192.88 | 0.89 ± 0.00 | 486646.40 ± 389587.7 |
| NLE | 0.87 ± 0.03 | 129.10 ± 38.56 | **0.90 ± 0.01** | **0.37 ± 0.39** |
| RCP | 0.87 ± 0.02 | 124.46 ± 29.45 | 0.90 ± 0.01 | 0.77 ± 0.87 |
| MCQR | 0.88 ± 0.03 | 1201.33 ± 802.06 | 0.89 ± 0.00 | 9149.48 ± 22.60 |
| Dist-Split | 0.55 ± 0.21 | 474.01 ± 273.14 | 0.02 ± 0.02 | 0.67 ± 1.32 |
| CQR | 0.91 ± 0.02 | 2093.45 ± 2367.83 | 0.91 ± 0.00 | 1096.55 ± 4508.45 |
| **JAPAN** | **0.90 ± 0.02** | **170.56 ± 74.59** | 0.90 ± 0.01 | 25.86 ± 31.53 |
| CopulaCPTS | 0.90 ± 0.03 | 252.95 ± 67.94 | 0.89 ± 0.01 | 11.41 ± 4.80 |
| VanilaCopula | 0.86 ± 0.04 | 175.12 ± 48.13 | 0.89 ± 0.01 | 0.74 ± 0.09 |
| CFRNN | 0.88 ± 0.02 | 207.45 ± 33.69 | 0.92 ± 0.01 | 1.34 ± 0.27 |
| JANET | 0.87 ± 0.04 | 207.44 ± 64.69 | 0.88 ± 0.01 | 6.37 ± 0.75 |

Table 5: **For Unconditional modelling**: Coverage and prediction volume for the **Energy** and **RF4D** datasets at $\epsilon = 0.10$. RF4D volumes are rescaled by $10^4$ for clarity. Methods with the lowest volume among those achieving coverage in $[0.89, 0.91]$ are bolded.

---

**Algorithm 2** JAPAN with Unconditional Normalising Flows

---

**Require:** Training set $\mathcal{D}_{\text{train}} = \{(x_i, y_i)\}_{i=1}^n$, calibration set $\mathcal{D}_{\text{cal}} = \{(x_j, y_j)\}_{j=1}^m$, significance level $\epsilon \in (0, 1)$
**Ensure:** Prediction set $\Gamma_\epsilon$ satisfying marginal coverage $\geq 1 - \epsilon$
1: Fit a normalising flow model to the marginal distribution of predictions $\hat{y}_i = \mu(x_i)$ using $\{\hat{y}_i\}_{i=1}^n$

2: **for** each calibration pair $(x_j, y_j) \in \mathcal{D}_{\text{cal}}$ **do**
3:     Compute conformity score: $\alpha_j = \log \hat{p}_{\hat{Y}}(y_j)$
4: **end for**
5: Compute threshold $\tau_\epsilon$ as the $k$-th smallest value in $\{\alpha_j\}_{j=1}^m$, where $k = \lfloor \epsilon \cdot (1 - 1/m) \cdot m \rfloor$
6: **At test time:** For any new input $x$, the prediction set using Proposition 3 such that:

$$\Gamma_\epsilon = \{y \in \mathcal{Y} \ : \ \log \hat{p}_{\hat{Y}}(y) \geq \tau_\epsilon\}$$

---

### A.3 CONDITIONAL MODELLING

We evaluate the conditional modelling extension of JAPAN on two tabular datasets: **Energy** and **RF4D**. In this setup, we first train a base predictive model $\mu$ to output point estimates $\hat{y} = \mu(x)$, and then train a conditional normalising flow to model $p(y \mid \hat{y})$. At inference, prediction regions are constructed by thresholding the log-likelihood of $y$ under the conditional flow (see Algorithm 3).

Table 6 summarises the coverage and areas for all methods. JAPAN consistently achieves competitive or lowest-volume prediction sets while maintaining valid coverage. On the **Energy** dataset, JAPAN achieves the most compact regions among all methods that fall within the target coverage range of $0.90 \pm 0.01$, outperforming all baselines in efficiency. In contrast, methods such as CQR, JANET, and CopulaCPTS yield significantly larger regions despite comparable coverage.

On the more challenging **RF4D** dataset, the benefits of conditional modelling are even more evident. JAPAN achieves a volume of just $0.69 \times 10^3$, nearly an order of magnitude smaller than most baselines, including residual-based and copula-based approaches. The flow model trained on $p(y \mid \hat{y})$ is able to adaptively reshape the predictive density to correct for errors in the base predictor, yielding highly compact and calibrated prediction areas. Notably, flow-free baselines such as MCQR, CFRNN, and CopulaCPTS result in excessive volumes or instability (e.g., MCQR reports region volumes on the order of $10^{17}$).

These results highlight the effectiveness of using JAPAN after a base model. Even though the flow is trained separately from the base model, its ability to correct for systematic biases in $\hat{y}$ enables significantly more efficient conformal sets than any competing method.

| Method | Cov. (Energy) | Area (Energy) | Cov. (RF4D) | Area (RF4D) × $10^3$ |
|---|---|---|---|---|
| CONTRA | 0.89 ± 0.03 | 53.87 ± 14.34 | 0.88 ± 0.01 | 2.28 ± 0.67 |
| PCP | 0.89 ± 0.02 | 176.66 ± 163.27 | 0.89 ± 0.01 | 8.94 ± 1.05 |
| NLE | 0.87 ± 0.04 | 82.75 ± 9.49 | 0.89 ± 0.00 | 91.46 ± 5.35 |
| RCP | 0.87 ± 0.02 | 124.46 ± 29.45 | 0.90 ± 0.01 | 77.15 ± 8.72 |
| MCQR | 0.87 ± 0.03 | 212.58 ± 145.09 | 0.89 ± 0.01 | 2.71e+17 ± 3.25e+17 |
| Dist-Split | 0.89 ± 0.03 | 151.26 ± 56.65 | 0.83 ± 0.01 | 20.69 ± 3.14 |
| CQR | 0.90 ± 0.02 | 232.93 ± 130.44 | 0.92 ± 0.00 | 20.75 ± 2.67 |
| **JAPAN** | **0.90 ± 0.01** | **50.10 ± 15.36** | **0.89 ± 0.01** | **0.69 ± 0.04** |
| CopulaCPTS | 0.89 ± 0.04 | 233.47 ± 69.06 | 0.89 ± 0.02 | 109.84 ± 28.84 |
| VanilaCopula | 0.86 ± 0.04 | 175.12 ± 48.13 | 0.92 ± 0.01 | 74.11 ± 9.46 |
| CFRNN | 0.88 ± 0.02 | 207.45 ± 33.69 | 0.92 ± 0.01 | 134.10 ± 27.16 |
| JANET | 0.90 ± 0.04 | 185.62 ± 38.79 | 0.89 ± 0.00 | 97.41 ± 28.28 |
| JAPAN Posterior | 0.88 ± 0.02 | 60.85 ± 5.23 | 0.89 ± 0.01 | 67.87 ± 16.40 |

Table 6: **For Conditional modelling:** Coverage and prediction volume for the **Energy** and **RF4D** datasets at $\alpha = 0.10$. RF4D volumes are rescaled by $10^3$ for clarity. Methods with the lowest volume among those achieving coverage in $[0.89, 0.91]$ are bolded. JAPAN has the lowest volume with the likelihood-based threshold, whereas JAPAN with the posterior threshold fares better than non-flow-based baselines.

---

**Algorithm 3** JAPAN with Conditional Normalising Flow on Predicted Inputs

---

**Require:** Training set $\mathcal{D}_{\text{train}} = \{(x_i, y_i)\}_{i=1}^n$, calibration set $\mathcal{D}_{\text{cal}} = \{(x_j, y_j)\}_{j=1}^m$, significance level
    $\epsilon \in (0, 1)$
**Ensure:** Prediction set $\Gamma_\epsilon(x)$ satisfying marginal coverage $\geq 1 - \epsilon$
 1: Train a base model $\mu$ on $\mathcal{D}_{\text{train}}$ to predict $\hat{y}_i = \mu(x_i)$
 2: Train a conditional normalising flow on pairs $(\hat{y}_i, y_i)$ to estimate $\hat{p}(y \mid \hat{y})$
 3: **for** each calibration point $(x_j, y_j) \in \mathcal{D}_{\text{cal}}$ **do**
 4:     Compute $\hat{y}_j = \mu(x_j)$
 5:     Compute conformity score: $\alpha_j = \log \hat{p}(y_j \mid \hat{y}_j)$
 6: **end for**
 7: Compute threshold $\tau_\epsilon$ as the $k$-th smallest value in $\{\alpha_j\}_{j=1}^m$, where $k = \lfloor \epsilon \cdot (1 - 1/m) \cdot m \rfloor$
 8: **At test time:** For any new input $x$, define the prediction set using Proposition 3 such that:
$$\Gamma_\epsilon(x) = \{y \in \mathcal{Y} \ : \ \log \hat{p}(y \mid \mu(x)) \geq \tau_\epsilon\}$$

---

### A.4 POSTERIOR MODELLING

We evaluate the posterior modelling extension of JAPAN on the **Energy** and **RF4D** datasets, using a normalising flow trained to model the inverse conditional density $p(\hat{y} \mid y)$, where $\hat{y} = \mu(x)$ is the output of a fixed base predictor. The posterior log-density is used as a conformity score to construct the calibrated prediction set as shown in Algorithm 4

As shown in Table 6, this approach achieves valid coverage on both datasets and produces reasonably compact prediction regions. On Energy, JAPAN Posterior yields a volume of 60.85, which is competitive with likelihood-based JAPAN and smaller than most non-flow baselines. On RF4D, however, the posterior-based method underperforms significantly against its likelihood counterpart, with a volume of 67.87 compared to 0.69 (rescaled), but remains more effective than most of the non-flow baselines, except for CQR and has a high coverage.

Intuitively, posterior-based conformity can be interpreted as testing how plausible an input $x$ (or a prediction $\hat{y}$) is given a candidate output $y$. This mirrors likelihood-based scoring in spirit: true $(x, y)$ pairs are expected to have high posterior values, and low-posterior values are assigned to implausible combinations. However, posterior calibration may not be practical in practice—particularly when the input dimensionality is high, and sampling becomes expensive (e.g., for area estimation), or when the posterior distribution $p(x \mid y)$ is more complex or less structured than the forward conditional

$p(y \mid x)$. In contrast, likelihood-based conformity scores allow efficient area estimation via latent-space sampling. (see Proposition 3)

Nevertheless, we re-emphasise that posterior-based JAPAN is a promising alternative in settings where conditional modelling is ill-posed (e.g., discrete $y$), or when the posterior distribution is available via a generative model.

In Figure 8, we visualise an example of a 2D Crescent-shaped density. Both methods achieve the desired coverage, but the posterior model yields a slightly smaller area (4.80 vs 5.57), suggesting potential gains in efficiency in some cases. However, the posterior region appears less symmetric, exhibiting stronger coverage along the upper edge of the density. It is still not clear if this asymmetrical nature can offer any advantages, or if there is any added theoretical value in using the posterior-based conformity score.

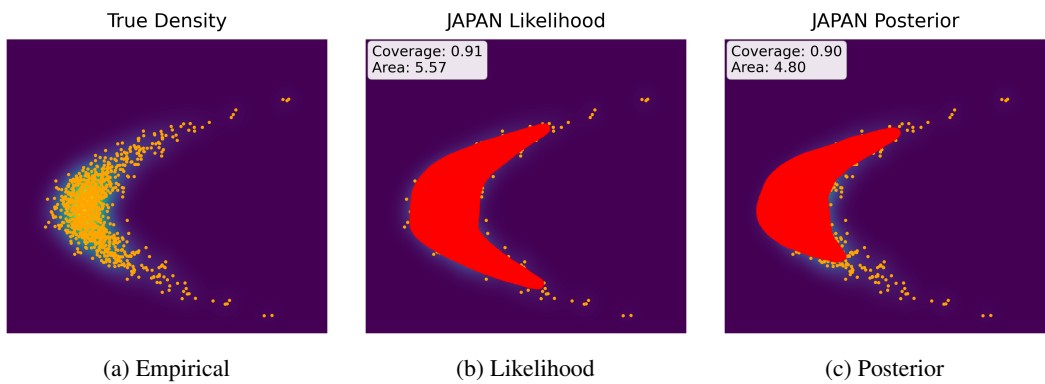

|  (a) Empirical  |  (b) Likelihood  |  (c) Posterior  |

Figure 8: Visualisation of JAPAN Likelihood vs JAPAN Posterior on Crescent density dataset at $\epsilon = 0.1$. Both achieve the desired coverage, but the Posterior density has a smaller region for the given example, while the Likelihood density looks more symmetric.

---

**Algorithm 4** JAPAN with Posterior Modelling ($p(x \mid y)$ or $p(\mu(x) \mid y)$)

---

**Require:** Training set $\mathcal{D}_{\text{train}} = \{(x_i, y_i)\}_{i=1}^n$, calibration set $\mathcal{D}_{\text{cal}} = \{(x_j, y_j)\}_{j=1}^m$, significance level $\epsilon \in (0, 1)$
**Ensure:** Prediction set $\Gamma_\epsilon(x)$ based on posterior log-likelihood conformity

1: **Option A (full input posterior):** Train a flow model $\hat{p}(x \mid y)$ using $\{(x_i, y_i)\}_{i=1}^n$
2: **Option B (prediction posterior):** Train a predictive model $\mu$, then fit $\hat{p}(\mu(x) \mid y)$ using $\{(\mu(x_i), y_i)\}_{i=1}^n$
3: **for** each calibration pair $(x_j, y_j) \in \mathcal{D}_{\text{cal}}$ **do**
4:     **For Option A:** set $\alpha_j = \log \hat{p}(x_j \mid y_j)$
5:     **For Option B:** compute $\hat{y}_j = \mu(x_j)$ and set $\alpha_j = \log \hat{p}(\hat{y}_j \mid y_j)$
6: **end for**
7: Compute threshold $\tau_\epsilon$ as the $k$-th smallest value in $\{\alpha_j\}_{j=1}^m$, where $k = \lfloor \epsilon \cdot (1 - 1/m) \cdot m \rfloor$
8: **At test time:** For any input $x$, define:

$$\Gamma_\epsilon(x) = \{y \in \mathcal{Y} \; : \; \log \hat{p}(x \mid y) \geq \tau_\epsilon\} \quad \text{or} \quad \log \hat{p}(\mu(x) \mid y) \geq \tau_\epsilon$$

depending on the variant used, and computed using Proposition 3.

---

## A.5 Latent Modelling

In the original JAPAN framework, the conformity score is computed with the full log-likelihood produced by a normalising flow as in Eq. (1):

$$\log p(y \mid x) = \log p_Z(h(y, x)) + \Phi(y, x).$$

where the first term accounted for the density of the transformed variable $z = h(y, x)$ in the latent distribution (e.g., standard Gaussian), and the second term accounted for the local change in area/volume under the flow transformation.

Here, we intentionally discard the area/volume correction term and define the conformity score using only the log-density of the transformed variable in the latent space (see Algorithm 5):

$$\alpha = \log p_Z(h(y, x)).$$

This corresponds to thresholding in the latent space without correcting for the local distortion introduced by the flow mapping. Intuitively, this results in a spherical region centred at the origin in latent space, which maps to an arbitrarily shaped region in data space. However, the region may inadvertently include low-density regions of the target distribution: low-density regions that were simply squeezed into high-density latent locations may be mistakenly assigned high conformity.

This design has a close connection to residual-based methods. The Gaussian base distribution naturally induces a conformity score that penalises distance from the centre, much like $\ell_2$-based residual scoring. Indeed, if the flow is volume-preserving (i.e., $\log |\det \partial h| = 0$) or if the area/volume change term is ignored, this formulation becomes equivalent to CONTRA (Fang et al., 2024), which bounds conformal balls in the latent space as their distance-based threshold corresponds to working with the density of transformed variable in the latent space. Hence, CONTRA can be viewed as a special case of JAPAN that uses only the transformed latent density, without accounting for how that density is warped back into data space. While this setup can produce valid regions, it often leads to large areas similar to CONTRA when modes are well-separated.

Formally, recall that $h$ denote the normalising-flow mapping conditioned on $x$, so that

$$z = h(y, x), \qquad z \sim p_Z,$$

where $p_Z$ is the base distribution. We choose $p_Z$ to be an isotropic Gaussian,

$$p_Z(z) = \frac{1}{(2\pi)^{d/2}} \exp\left(-\frac{1}{2}\|z\|_2^2\right),$$

so its log-density is

$$\log p_Z(z) = -\frac{1}{2}\|z\|_2^2 - \frac{d}{2}\log(2\pi).$$

CONTRA's conformity score (latent-ball version) can be written as

$$\alpha_{\text{CONTRA}}(y \mid x) = -\|h(y, x)\|_2.$$

Note that we change the original non-conformity score to conformity score for the comparison by changing the sign.

JAPAN's latent-density score is:

$$\alpha_{\text{JAPAN}}(y \mid x) = \log p_Z(h(y, x)) = -\frac{1}{2}\|h(y, x)\|_2^2 - \frac{d}{2}\log(2\pi).$$

Then for any $y$ and $x$, we have

$$\alpha_{\text{JAPAN}}(y \mid x) = \alpha_{\text{CONTRA}}(y \mid x) + \text{constant}.$$

Crucially, for any two candidate outputs $y_1, y_2$ (and fixed $x$),

$$\alpha_{\text{CONTRA}}(y_1 \mid x) \leq \alpha_{\text{CONTRA}}(y_2 \mid x) \iff \alpha_{\text{JAPAN}}(y_1 \mid x) \geq \alpha_{\text{JAPAN}}(y_2 \mid x).$$

That is, the Euclidean norm in latent space and the latent Gaussian log-density induce **exactly the same ranking** over candidate $y$'s and equivalently the same prediction regions/areas.

In Figure 9, we show comparisons of different variants on JAPAN with visualisation of areas computed at $\epsilon = 0.1$ for the Crescent-shaped density. Importantly, the coverage and the computed areas for CONTRA and JAPAN-Latent are identical, showing empirically that CONTRA is a special case within JAPAN.

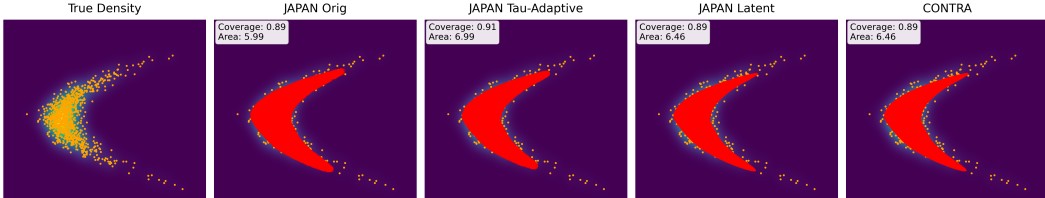

Figure 9: **Visualising prediction regions across JAPAN variants.** From left to right: the true density, **JAPAN Original**, **JAPAN Tau-Adaptive**, **JAPAN Latent**, and **CONTRA**. All methods achieve comparable marginal coverage (near $0.89$), but the shape and compactness of the prediction regions vary. JAPAN Orig applies a fixed threshold across contexts, while Tau-Adaptive adjusts the threshold locally to account for varying volume distortion. JAPAN-Latent and CONTRA both threshold in latent space, resulting in same areas and coverage showing JAPAN in latent is equivalent to CONTRA. Tau-Adaptive yields improved coverage and competitive area, closely matching the data geometry.

---

**Algorithm 5** JAPAN with Latent Modelling

---

**Require:** Training set $\mathcal{D}_{\text{train}} = \{(x_i, y_i)\}_{i=1}^n$, calibration set $\mathcal{D}_{\text{cal}} = \{(x_j, y_j)\}_{j=1}^m$, significance level $\epsilon \in (0, 1)$
**Ensure:** Prediction set $\Gamma_\epsilon(x)$ using latent-space density as conformity score
1: Train a conditional normalising flow $h(y, x)$, where $z = h(y, x)$ and $z \sim p_Z$
2: **for** each calibration pair $(x_j, y_j)$ **do**
3:    Compute latent variable: $z_j = h(y_j, x_j)$
4:    Compute conformity score: $\alpha_j = \log p_Z(z_j)$
5: **end for**
6: Compute threshold $\tau_\epsilon$ as the $k$-th smallest value in $\{\alpha_j\}_{j=1}^m$, where $k = \lfloor \epsilon \cdot (1 - 1/m) \cdot m \rfloor$
7: **At test time:** For input $x$, define:
$$\Gamma_\epsilon(x) = \{y \in \mathcal{Y} \ : \ \log p_Z(h(y, x)) \geq \tau_\epsilon\}$$

---

## A.6   ADAPTIVE $\tau_\epsilon(x)$

While JAPAN produces context-adaptive regions and supports disjoint prediction sets, it does not guarantee *conditional coverage*. This limitation arises because the conformal threshold $\tau_\epsilon$ is computed globally, independent of the conditioning variable $x$. As a result, although the shape of the region adapts with $x$, its size may be suboptimal or overly conservative in certain regions of the input space.

To address this, we propose a *tau-adaptive* extension of JAPAN, in which the threshold itself is a function of $x$, i.e., $\tau_\epsilon(x)$. Our approach builds on the fact that the log-likelihood in a normalising flow decomposes into two terms, from Eq. (1):

$$\log p(y \mid x) = \underbrace{\log p_Z(h(y, x))}_{\text{latent density}} + \underbrace{\Phi(y, x)}_{\text{area change}} .$$

We use this decomposition to split the conformity score into two components. The **unconditional component** is based on the latent density and is calibrated globally using the standard conformal approach. This yields a latent-space threshold, which corresponds to the same threshold used in our latent modelling extension Algorithm 5 or CONTRA Fang et al. (2024). Formally, we compute the global latent-space threshold $\tilde{\tau}_\epsilon$ from the calibration scores $\log p_Z(h(y_j, x_j))$, and additionally store the corresponding latent vector $z_\epsilon$ for the $\epsilon$-quantile (i.e., the $k$-th smallest score). At test time, we use this fixed latent value to construct a context-specific threshold in the data space:

$$\tau_\epsilon(x) = \tilde{\tau}_\epsilon + \Phi(h^{-1}(z_\epsilon, x), x).$$

This formulation ensures that the test-time threshold accounts for local flow transformations by evaluating how the calibration-latent $z_\epsilon$ maps back to data space under the test-time context $x$. Further details on the algorithm are in Algorithm 6.

Intuitively, this adjustment of the threshold using the **area-change component** reflects how the flow expands or contracts volume around $x$. When the Jacobian suggests strong expansion (i.e., high uncertainty), the threshold is lowered, allowing a larger region; conversely, in contracting regions (i.e., high confidence), the threshold is raised. It can also be viewed as aligning the latent-space decision boundary with the true conditional-density contour in the data space, and thus restoring a degree of *conditional coverage*. We may even get nearly optimal prediction regions under certain assumptions (e.g., when $p(y \mid x)$ transforms linearly in $x$ and thus maps the true thresholds to the same latent vector, and the flow estimates conditional density sufficiently well across $x$). It improves upon other context-adaptive methods, which only modulate shape by shifting and not scaling, and implicitly assume area-preserving transformations. An alternative to using the latent vector obtained through global calibration can be to find the local threshold using the neighbours of the context, which also brings JAPAN closer to conditional coverage or more specifically, local coverage. Algorithm 7 shows the full algorithm for this proposal.

In Figure 9, we show the prediction regions obtained by the tau-adaptive extension on the Crescent dataset. While the resulting region is slightly less compact than those produced by CONTRA or other JAPAN extensions (with area 6.99 compared to 6.46 and 6.46, 5.99, respectively), it achieves slightly improved coverage (0.91), reflecting the influence of threshold modulation through local volume expansion. This suggests that adapting the threshold based on flow geometry can provide a useful mechanism for refining predictive regions, even if it may not always lead to the smallest volume. Unconditional coverages can easily form tighter regions by excluding high-variance predictions, but this comes at the cost of non-uniform calibration conditioned on context. Although the tau-adaptive variant introduces a modest increase in area, it may offer a practical pathway toward better conditional coverage.

**Illustration on a Toy Example:** To further highlight the role of the tau-adaptive extension, we design a simple heteroscedastic regression problem. Let

$$x \sim \mathcal{U}[0,1], \qquad y = ax + x \cdot \epsilon, \qquad \epsilon \sim \mathcal{N}(0,1),$$

so that the conditional distribution is

$$y \mid x \sim \mathcal{N}(ax, x^2).$$

This setting induces a conditional variance that increases with $x$, making it a natural testbed for evaluating conditional coverage. Since the latent mapping is fixed across $x$, the tau-adaptive mechanism can be applied analytically, and the adaptive threshold $\tau_\epsilon(x)$ adjusts to the local variance.

As shown in Table 7, JAPAN with tau-adaptive thresholding achieves the desired conditional coverage (0.90) while also adapting the prediction region width appropriately: smaller when $x = 0.5$ and larger when $x = 1$, consistent with the variance structure of the model.

Table 7: The conditional coverage for a toy example, where our assumptions (latent mapping and hence $\tau$ is the same for different values of $x$) are satisfied.

| Method | Width | Coverage |
|---|---|---|
| JAPAN ($x = 1$) | $3.99 \pm 0.01$ | 0.90 |
| JAPAN ($x = 0.5$) | $1.64 \pm 0.01$ | 0.90 |

Since our primary goal is to highlight the flexibility of flow-based conformity scoring, we defer a deeper investigation into conditional coverage guarantees to future work.

A.7 ABLATION: EFFECT OF UNDERFITTING IN FLOW MODELS

We begin by recalling the role of conformity scores in our framework. In traditional inductive conformal prediction, the predictor $\mu$ and the scoring rule $\mathcal{A}$ are defined separately, e.g. using residual-based scores such as $\alpha_i = -|y_i - \mu(x_i)|$. In contrast, JAPAN has multiple settings, for e.g the first setting as in Section 3.3 of JAPAN unifies these components: the normalising flow is not only used to obtain a predictive model, but also directly provides the conformity score via its estimated log-likelihood, $\log \hat{p}(y_i \mid x_i)$. This removes the need for a separate residual-based metric

---

**Algorithm 6** $\tau$-Adaptive JAPAN

---

**Require:** Training set $\mathcal{D}_{\text{train}} = \{(x_i, y_i)\}_{i=1}^n$, calibration set $\mathcal{D}_{\text{cal}} = \{(x_j, y_j)\}_{j=1}^m$, significance level $\epsilon \in (0, 1)$
**Ensure:** Prediction set $\Gamma_\epsilon(x)$ with geometry-aware adaptive thresholding
 1: Train a conditional normalising flow: $z = h(y, x)$, with base density $p_Z$
 2: **for** each calibration pair $(x_j, y_j)$ **do**
 3:    Compute latent representation $z_j = h(y_j, x_j)$
 4:    Set conformity score: $\alpha_j = \log p_Z(z_j)$
 5: **end for**
 6: Compute global latent threshold $\tilde{\tau}_\epsilon$ as the $k$-th smallest value in $\{\alpha_j\}_{j=1}^m$, where $k = \lfloor \epsilon \cdot (1 - 1/m) \cdot m \rfloor$ and store the corresponding latent vector $z_\epsilon$.
 7: **At test time:** For input $x$, define the data-space threshold:

$$\tau_\epsilon(x) = \tilde{\tau}_\epsilon + \Phi\left(h^{-1}(z_\epsilon, x), x\right)$$

 8: Predictive region:

$$\Gamma_\epsilon(x) = \{y \in \mathcal{Y} \ : \ \log p_Z(h(y, x)) + \Phi(y, x) \geq \tau_\epsilon(x)\}$$

---

**Algorithm 7** $\tau_\epsilon(x)$-Adaptive JAPAN with Neighbour-based local adaptivity

---

**Require:** Training set $\mathcal{D}_{\text{train}} = \{(x_i, y_i)\}_{i=1}^n$, calibration set $\mathcal{D}_{\text{cal}} = \{(x_j, y_j)\}_{j=1}^m$, significance level $\epsilon \in (0, 1)$
**Ensure:** Prediction set $\Gamma_\epsilon(x)$ with locally adaptive thresholding
 1: Train a conditional normalising flow: $z = h(y, x)$, with base density $p_Z$

 2: **for** each calibration pair $(x_j, y_j) \in \mathcal{D}_{\text{cal}}$ **do**
 3:    Compute latent variable $z_j = h(y_j, x_j)$
 4:    Store: latent conformity $\alpha_j = \log p_Z(z_j)$, correction term $\Phi_j = \Phi(y_j, x_j)$
 5: **end for**

 6: Compute global latent threshold $\tilde{\tau}_\epsilon$ as the $k$-th smallest in $\{\alpha_j\}_{j=1}^m$, where $k = \lfloor \epsilon \cdot (1 - 1/m) \cdot m \rfloor$

 7: **At test time:** For input $x$:
   - Compute $\hat{\Phi}(x) \approx$ mean of $\Phi_j$ over the $K$-nearest calibration inputs $x_j$
   - Set threshold $\tau_\epsilon(x) = \tilde{\tau}_\epsilon + \hat{\Phi}(x)$
   - Define:
$$\Gamma_\epsilon(x) = \{y \in \mathcal{Y} \ : \ \log p_Z(h(y, x)) + \Phi(y, x) \geq \tau_\epsilon(x)\}$$

---

and ties prediction and conformity into a single probabilistic model, whereas the second setting, as in Algorithm 3, used a separate base predictor.

A natural question is how the performance of JAPAN degrades when the underlying flow is underfitted, particularly in high-dimensional settings. While coverage guarantees remain valid regardless of the flow quality, the efficiency of the prediction regions, i.e., their area, may degrade. The extent of this degradation depends on how poorly the flow model approximates the true distribution. Typically, normalising flows trained via maximum likelihood tend to assign higher density to frequently observed samples, which tends to preserve ranking structures reasonably well even under underfitting.

We first revisit the setting where both predictor $\mu$ and conformity function $\mathcal{A}$ are defined jointly by the normalising flow. The central question here is whether the prediction regions degrade if the normalising flow is not well-trained. The answer parallels what happens in standard conformal prediction when the predictor $\mu$ is underfitted: in both cases, a poorly trained model results in unreliable conformity scores and hence inefficient or wide prediction sets. Poor flow training is a limitation, but it only widens the prediction sets to account for that uncertainty, which is inherent to all conformal methods.

In fact, all of our main baselines also use normalising flows (for the mean predictor), but rely on residuals or geometric heuristics for scoring. If the flow model is inaccurate in those cases, the reliability of the prediction regions also deteriorates. In this ablation, where we reduce the number of layers to half, we see in Table 8 that JAPAN still achieved more efficient sets compared to baselines, even though the flow was underfitting. We posit this is because the model still learns reasonably well to put high density on the frequently occurring outcomes.

Table 8: Underfitted modelling: Coverage and prediction region area on the Energy and RF4D datasets. Lower area indicates better efficiency at the same coverage.

| Method | Coverage (Energy) | Area (Energy) | Coverage (RF4D) | Area (RF4D) |
|---|---|---|---|---|
| JAPAN | **0.89 ± 0.01** | **25.61 ± 2.44** | **0.90 ± 0.01** | **407.68 ± 44.19** |
| CONTRA | 0.89 ± 0.01 | 28.13 ± 6.18 | 0.90 ± 0.01 | 690.99 ± 108.35 |
| CopulaCPTS | 0.88 ± 0.01 | 40.63 ± 12.33 | 0.91 ± 0.01 | 2991.77 ± 5123.00 |

Importantly, in this setting, the conformity function $\mathcal{A}$ and the predictive component $\mu$ were jointly estimated by the normalising flow. This coupling makes it difficult to isolate underfitting effects on $\mathcal{A}$ or $\mu$ individually. In contrast, most baseline methods relied on fixed predictors and separately defined conformity scores, simplifying such analysis.

To further investigate this, we draw attention to the second setting, i.e. the experimental setting in Table 6, which decouples the predictive mean from the conformity score. The results showed that applying the normalising flow improves uncertainty estimation of the base model. We follow the same setting here, and purposely underfit the normalising flow. Note that the normalising flows are only used here in JAPAN, and thus the prediction of other models (baselines) remains constant from Table 6. In Table 9, we see that JAPAN still yielded much better results, even though it was underfitting with only half the layers used for training.

Table 9: Underfitted modelling after decoupling scoring and predictor model: Coverage and prediction region area on the Energy and RF4D datasets. We chose two non-flow-based baselines that worked the best in this setting. Lower area indicates better efficiency at the same coverage.

| Method | Coverage (Energy) | Area (Energy) | Coverage (RF4D) | Area (RF4D) $\times 1000$ |
|---|---|---|---|---|
| JAPAN | **0.90 ± 0.01** | **76.19 ± 4.92** | **0.89 ± 0.01** | **9.91 ± 1.12** |
| Dist-Split | 0.89 ± 0.03 | 151.26 ± 56.65 | 0.83 ± 0.01 | 20.69 ± 3.14 |
| RCP | 0.87 ± 0.02 | 124.46 ± 29.45 | 0.90 ± 0.01 | 77.15 ± 8.72 |

In summary, while underfitting reduces efficiency across all conformal methods, JAPAN degrades less. Because the log-likelihood estimated by the flow simultaneously governs both prediction and conformity, JAPAN remains competitive even when the flow is not fully expressive.

Table 10: Samples required for comparable accuracy on the test set with sampling schemes sampling, $d = 4$ and $d = 8$.

| Method | Samples ($d = 4$) | Time (minutes) | Samples ($d = 8$) | Time (minutes) |
|---|---|---|---|---|
| Proposition 3 (ours) | **1,000** | **0.4** | **3,000** | **2.5** |
| Naive MC | 100,000 | 5 | 1,000,000 | 32 |

## B  PROOFS

**Proposition 1** [Optimality Under Rank Preservation]

Let $p(y \mid x)$ denote the true conditional density, and let $f_\theta(y \mid x)$ denote a model-estimated conditional density. Consider the prediction region defined at coverage level $1 - \epsilon$:

$$\Gamma_\epsilon^*(x) = \{y : p(y \mid x) \geq \tau^*\}, \quad \text{where} \quad \mathbb{P}(y \in \Gamma_\epsilon^*(x)) = 1 - \epsilon,$$

and the conformal prediction region defined via the model as:

$$\Gamma_\epsilon(x) = \{y : f_\theta(y \mid x) \geq \tau\}, \quad \text{where} \quad \mathbb{P}(y \in \Gamma_\epsilon(x)) = 1 - \epsilon.$$

Assume:

- There exists a strictly monotone increasing function $g : \mathbb{R}_+ \to \mathbb{R}_+$ such that
$$f_\theta(y \mid x) = g(p(y \mid x)), \quad \forall y, x,$$
- The conditional densities $p(y \mid x)$ are **homogeneous across** $x$, i.e., there exists a base density $\phi$ and a translative transformation $T_x$ such that
$$p(y \mid x) = \phi(T_x(y)) \quad \text{for all } x.$$

Then the model-defined conformal region achieves the same area as the true optimal region:

$$\text{Area}(\Gamma_\epsilon(x)) = \text{Area}(\Gamma_\epsilon^*(x)) \quad \text{for all } x.$$

**Remark:** The homogenity condition assume that the **shape** of the conditional densities $p(y \mid x)$ remains the same across different values of $x$, up to translation based transformations. More generally, this means that the conditional distributions share the same spread or covariance structure but may differ in location. This assumption is standard in settings where the conditional variability is constant but the conditional mean changes. Examples include Gaussian location families with fixed covariance, or conditional distributions that differ only by a shift parameter.

*Proof.* We know that the prediction region at coverage level $1 - \epsilon$ under $p(y \mid x)$ is:

$$\Gamma_\epsilon^*(x) = \{y : p(y \mid x) \geq \tau^*\}, \quad \text{with} \quad \int_{\Gamma_\epsilon^*(x)} p(y \mid x)\, dy = 1 - \epsilon.$$

By assumption, $f_\theta(y \mid x) = g(p(y \mid x))$ for a strictly increasing function $g$. Hence, its inverse $g^{-1}$ is also strictly increasing, and:

$$p(y \mid x) = g^{-1}(f_\theta(y \mid x)).$$

Then the model-based conformal region is:

$$\Gamma_\epsilon(x) = \{y : f_\theta(y \mid x) \geq \tau\} = \{y : p(y \mid x) \geq g^{-1}(\tau)\}.$$

Since both regions achieve the same coverage level under the same true distribution $p(y \mid x)$, they must correspond to the same threshold:

$$g^{-1}(\tau) = \tau^* \quad \Longleftrightarrow \quad \tau = g(\tau^*),$$

implying:

$$\Gamma_\epsilon(x) = \Gamma_\epsilon^*(x).$$

Finally, under the homogeneity assumption, the structure of the density and its level sets is invariant (up to transformation) across different values of $x$. Therefore, area computations under the transformation $T_x$ preserve the equality:

$$\text{Area}(\Gamma_\epsilon(x)) = \text{Area}(\Gamma_\epsilon^*(x)).$$

$\square$

**Structural Equivalence Under Monotonic Transformation:** Even without assuming homogeneity of the conditional densities $p(y \mid x)$, if the model density $f_\theta(y \mid x)$ is a strictly monotonic transformation of the true density (i.e., $f_\theta(y \mid x) = g(p(y \mid x))$ for some fixed, monotone $g$), then the prediction regions defined by thresholding either density are identical for each $x$, and hence their expected areas match:

$$\Gamma_\epsilon(x) = \Gamma_\epsilon^*(x) \quad \implies \quad \mathbb{E}_x[\text{Area}(\Gamma_\epsilon(x))] = \mathbb{E}_x[\text{Area}(\Gamma_\epsilon^*(x))].$$

This does not imply that the model-based region is optimal, but rather that it is *structurally equivalent* to the one derived from the true density under the same thresholding scheme.

*Implication for JAPAN:* Since the logarithm is a strictly monotonic transformation, this proposition also justifies using log-likelihood (i.e., $\log \hat{p}(y \mid x)$) as a conformity score in JAPAN, without loss of structural validity in the regions.

**Proposition 2** [Approximate Optimality under Approximate Rank Preservation] Let $p(y \mid x)$ be the true conditional density and $f_\theta(y \mid x)$ an estimated conditional density. Suppose there exists a strictly monotone increasing function $g : \mathbb{R}_+ \to \mathbb{R}_+$ and a positive constant $\delta$ such that for all $y$ and $x$:

$$|f_\theta(y \mid x) - g(p(y \mid x))| \leq \delta.$$

Assume further that the conditional densities $p(y \mid x)$ are **homogeneous across** $x$, i.e., there exists a base density $\phi$ and a translative transformation $T_x$ such that:

$$p(y \mid x) = \phi(T_x(y)) \quad \text{for all } x.$$

Define the optimal and estimated prediction regions respectively as:

$$\Gamma_\epsilon^*(x) = \{y : p(y \mid x) \geq \tau^*\}, \quad \Gamma_\epsilon(x) = \{y : f_\theta(y \mid x) \geq \tau\},$$

both satisfying coverage:

$$\mathbb{P}(y \in \Gamma_\epsilon^*(x)) = \mathbb{P}(y \in \Gamma_\epsilon(x)) = 1 - \epsilon.$$

Then the area difference is bounded and approaches zero as $\delta \to 0$:

$$|\text{Area}(\Gamma_\epsilon(x)) - \text{Area}(\Gamma_\epsilon^*(x))| \leq C(\delta), \quad \text{with } C(\delta) \to 0 \text{ as } \delta \to 0.$$

*Proof.* By definition of coverage, we have:

$$\int_{\Gamma_\epsilon^*(x)} p(y \mid x) \, dy = 1 - \epsilon, \quad \text{and} \quad \int_{\Gamma_\epsilon(x)} p(y \mid x) \, dy = 1 - \epsilon.$$

Since $f_\theta(y \mid x)$ approximates $g(p(y \mid x))$ uniformly within $\delta$, we have:

$$\Gamma_\epsilon(x) = \{y : f_\theta(y \mid x) \geq \tau\} \approx \{y : g(p(y \mid x)) \geq \tau\}.$$

Thus, the thresholds $\tau^*$ and $g^{-1}(\tau)$ must be close. Formally, continuity and strict monotonicity of $g$ and $g^{-1}$ imply:

$$|\tau^* - g^{-1}(\tau)| \leq \varepsilon(\delta),$$

with $\varepsilon(\delta) \to 0$ as $\delta \to 0$.

Consider the symmetric difference of the two regions:

$$\Gamma_\epsilon^*(x) \triangle \Gamma_\epsilon(x) = \{y : p(y \mid x) \in [\min(\tau^*, g^{-1}(\tau)), \max(\tau^*, g^{-1}(\tau))]\}.$$

This set corresponds to a thin boundary region around level sets defined by $p(y \mid x)$, with width at most $\varepsilon(\delta)$.

The area difference is bounded by the area of this boundary set:

$$|\text{Area}(\Gamma_\epsilon(x)) - \text{Area}(\Gamma_\epsilon^*(x))| \leq \text{Area}(\Gamma_\epsilon^*(x) \triangle \Gamma_\epsilon(x)).$$

By the homogeneity assumption, the shape and structure of level sets across different $x$ are invariant (up to transformation), so the area difference remains controlled uniformly. Hence:

$$\text{Area}(\Gamma_\epsilon^*(x) \triangle \Gamma_\epsilon(x)) \leq C(\delta), \quad \text{with } C(\delta) \to 0 \text{ as } \delta \to 0.$$

$\square$

**Proposition 3** [Area (or Volume) Computation] Let $z \sim p_Z(z)$ be samples from a base distribution, and let $y = h^{-1}(z, x)$ denote the inverse flow transformation conditioned on $x$, and $\exp(\phi(z, x))$ denote the area/volume correction under the inverse transformation. Then the area (or volume) of the conformal prediction region $\Gamma_\epsilon(x)$ can be approximated as:

$$\widehat{\mathrm{Area}}(\Gamma_\epsilon(x)) = \frac{1}{N} \sum_{i=1}^N \mathbb{1} \left\{ \hat{p}(h^{-1}(z_i, x) \mid x) \geq \tau_\epsilon \right\} \cdot \frac{\exp(\phi(z, x))}{p_Z(z_i)},$$

where $z_i \sim p_Z$ for $i = 1, \ldots, N$, and $\tau_\epsilon$ is the conformal threshold.

*Proof.* Let $F(z) := \mathbb{1} \left\{ \hat{p}(h^{-1}(z, x) \mid x) \geq \tau_\epsilon \right\} \cdot \exp(\phi(z, x))$. Then the area of the conformal region can be written as an integral over the data space:

$$\mathrm{Area}(\Gamma_\epsilon(x)) = \int_{\mathbb{R}^{D_y}} \mathbb{1} \left\{ \hat{p}(y \mid x) \geq \tau_\epsilon \right\} dy.$$

Changing variables via the inverse flow $y = h^{-1}(z, x)$, we obtain:

$$\mathrm{Area}(\Gamma_\epsilon(x)) = \int_{\mathbb{R}^{D_z}} F(z) \, dz.$$

Expressing this as an expectation under $z \sim p_Z(z)$, we get:

$$\mathrm{Area}(\Gamma_\epsilon(x)) = \mathbb{E}_{z \sim p_Z} \left[ \frac{F(z)}{p_Z(z)} \right] = \mathbb{E}_{z \sim p_Z} \left[ \mathbb{1} \left\{ \hat{p}(h^{-1}(z, x) \mid x) \geq \tau \right\} \cdot \frac{\exp(\phi(z, x))}{p_Z(z)} \right].$$

This expectation can be estimated with $N$ Monte Carlo samples $\{z_i \sim p_Z\}_{i=1}^N$, yielding:

$$\widehat{\mathrm{Area}}(\Gamma_\epsilon(x)) = \frac{1}{N} \sum_{i=1}^N \mathbb{1} \left\{ \hat{p}(h^{-1}(z_i, x) \mid x) \geq \tau \right\} \cdot \frac{\exp(\phi(z, x))}{p_Z(z_i)}.$$

This estimate accounts for area/volume transformation under $h^{-1}$, and evaluates only those points whose mapped outputs lie within the conformal region. When using discrete Normalising Flows, we can write down $\exp(\phi(z, x))$ as $|\det J_{h^{-1}}(Z, x)|$, where $J_{h^{-1}}$ is the Jacobian under the inverse transformation. Then, the computed area takes the form:

$$\widehat{\mathrm{Area}}(\Gamma_\epsilon(x)) = \frac{1}{N} \sum_{i=1}^N \mathbb{1} \left\{ \hat{p}(h^{-1}(z_i, x) \mid x) \geq \tau \right\} \cdot \frac{|\det J_{h^{-1}}(z_i, x)|}{p_Z(z_i)}.$$

$\square$

## B.1 Alternative view of Volume/Area estimator as Importance Sampling

Whilst sampling from the latent space is effectively equivalent to sampling from the data distribution, and contributes to the efficiency of the estimation process. We can also view the volume/area estimator as an importance sampling estimator, which helps explain both its correctness and practical efficiency.

Using the change-of-variables formula, the volume of the prediction region

$$\Gamma_\epsilon(x) = \{y : \log f_\theta(y \mid x) \geq \tau\}$$

can be written as

$$\mathrm{Area}(\Gamma_\epsilon(x)) = \int_{h(\Gamma_\epsilon(x))} |\det J_{h^{-1}}(z, x)| \, dz.$$

Here, $h^{-1}$ is the transformation from the latent space to the data space, and $J_{h^{-1}}$ is the Jacobian of the transformation.

By introducing the latent Gaussian $p_Z(z)$ and multiplying and dividing by $p_Z(z)$, this becomes the expectation

$$\text{Area}(\Gamma_\epsilon(x)) = E_{z \sim p_Z}\left[\frac{|\det J_{h^{-1}}(z,x)|}{p_Z(z)}\mathbf{1}(h^{-1}(z,x) \in \Gamma_\epsilon(x))\right],$$

which is exactly the form of an **importance sampling estimator** with proposal distribution $p_Z$. The weights

$$w(z,x) = \frac{|\det J_{h^{-1}}(z,x)|}{p_Z(z)}$$

are the corresponding importance weights, and the indicator selects points in the prediction region. Our Monte Carlo estimator is therefore an Importance Sampling estimator:

$$\widehat{\text{Area}}(\Gamma_\epsilon(x)) = \frac{1}{M}\sum_{i=1}^{M} w(z_i, x)\,\mathbf{1}\left(h^{-1}(z_i, x) \in \Gamma_\epsilon(x)\right), \qquad z_i \sim p_Z.$$

This Importance Sampling viewpoint also explains the **efficiency** of the estimator. Because normalising flows ,typically, map the data distribution to Gaussian latent $p_Z$, and the data variable $y$ can be expressed as a function of $z$, the Importance Sampling viewpoint of using the Gaussian as the proposal distribution. This avoids the inefficiency of naive grid-sampling or uniform sampling in the output space. In other words, the proposal distribution $p_Z$ is well-aligned with the transformation-induced distribution of the data via flows, which also reduces the **variance of the estimator**, as the weights are naturally obtained from the volume change component that can be analytically calculated via flows.

**Proposition 4** (Bounded Misranking Leads to Bounded Excess Area). *Let $p(y \mid x)$ denote the true conditional density, and let $f_\theta(y \mid x)$ denote the estimated conditional density from a model. Define the conformal prediction region and the optimal region at coverage level $1 - \epsilon$ as:*

$$\Gamma_\epsilon(x) = \{y : f_\theta(y \mid x) \geq \tau\}, \quad \Gamma_\epsilon^*(x) = \{y : p(y \mid x) \geq \tau^*\},$$

*with thresholds $\tau$ and $\tau^*$ chosen so that:*

$$\mathbb{P}(y \in \Gamma_\epsilon(x)) = \mathbb{P}(y \in \Gamma_\epsilon^*(x)) = 1 - \epsilon.$$

*We use $\text{Area}(\cdot)$ to denote the Lebesgue measure of a region in $\mathbb{R}^{D_y}$.*

*Define the* misranking probability mass*:*

$$\mu(x) = \iint_{\mathcal{M}_x} p(y_1 \mid x)\,p(y_2 \mid x)\,dy_1\,dy_2, \quad \text{where} \quad \mathcal{M}_x = \{(y_1, y_2) : p(y_1 \mid x) > p(y_2 \mid x),\, f_\theta(y_1 \mid x) < f_\theta(y_2 \mid x)\}.$$

*Then there exists a function $C(\cdot)$ with $C(\mu) \to 0$ as $\mu \to 0$ such that:*

$$|\text{Area}(\Gamma_\epsilon(x)) - \text{Area}(\Gamma_\epsilon^*(x))| \leq C(\mu(x)).$$

*Proof.* Let us consider the symmetric difference between the two regions:

$$\Gamma_\epsilon(x) \triangle \Gamma_\epsilon^*(x) = (\Gamma_\epsilon(x) \setminus \Gamma_\epsilon^*(x)) \cup (\Gamma_\epsilon^*(x) \setminus \Gamma_\epsilon(x)).$$

Any point $y$ in this symmetric difference satisfies:

$$(p(y \mid x) - \tau^*)(f_\theta(y \mid x) - \tau) < 0,$$

which means that $y$ lies above one threshold and below the other — precisely due to a ranking discrepancy between $f_\theta$ and $p$ near the threshold boundary.

Now consider the set of all such $y$ for which this misalignment occurs. These are generated by misranked pairs — i.e., pairs $(y_1, y_2)$ such that $p(y_1 \mid x) > p(y_2 \mid x)$ but $f_\theta(y_1 \mid x) < f_\theta(y_2 \mid x)$. The total measure of such misranked pairs is quantified by $\mu(x)$.

When $\mu(x)$ is small, the threshold $\tau$ will be close to $g(\tau^*)$ (assuming $f_\theta$ approximates a monotonic transformation of $p$), and the region boundaries will be close. This implies that the symmetric

difference between $\Gamma_\epsilon(x)$ and $\Gamma_\epsilon^*(x)$ is thin — geometrically, it lies in a narrow strip of points near the threshold.

Therefore, the area of the symmetric difference is bounded by a function $C(\mu(x))$, which goes to zero as $\mu(x) \to 0$:

$$\text{Area}(\Gamma_\epsilon(x) \triangle \Gamma_\epsilon^*(x)) \leq C(\mu(x)).$$

Since:

$$|\text{Area}(\Gamma_\epsilon(x)) - \text{Area}(\Gamma_\epsilon^*(x))| \leq \text{Area}(\Gamma_\epsilon(x) \triangle \Gamma_\epsilon^*(x)),$$

we conclude the result. $\qquad\qquad\square$

## C  EXPERIMENTAL AND ARCHITECTURAL DETAILS

### C.1  TARFLOW FOR TIME SERIES MODELLING

We adapt the TARFLOW architecture (Zhai et al., 2024), originally proposed for image patch modelling, to the time series setting by treating the forecast window as a sequence of tokens and conditioning the flow on the past context. Each time series forecast target $y \in \mathbb{R}^{H \times d}$ is modelled autoregressively using a stack of Transformer-based flow blocks. These blocks, as in TARFLOW, contain a causal Transformer followed by diagonal affine transformations applied to permuted inputs. Unlike the image setting, where the input is a sequence of visual patches, we treat the temporal axis as the natural ordering and apply causal masking accordingly.

To incorporate conditioning on the historical context $x \in \mathbb{R}^{T \times d}$, we introduce an additive conditioning path. Specifically, we project the input $x$ through a context encoder and map it to a sequence $c \in \mathbb{R}^{H \times d}$ of the same length and shape as the target $y$. This projected context is then added elementwise to the latent representations inside the Transformer flow block:

$$z^{l+1} = f^l(z^l) + \text{context encoder}(x),$$

where $f^l$ is the Transformer autoregressive block, $l$ is the layer index, and context encoder$(x)$ denotes the conditioning signal projected to match the forecast window.

Briefly, each flow block $f^l$ follows a block autoregressive architecture. The core operation consists of three components:

1. **Causal Transformer:** A standard Transformer with causal masking (along the time axis) processes $z^l$ to produce context-based representations. The attention mechanism ensures each timestep $z_i^l$ can only attend to past and current positions.
2. **Diagonal Affine Transform:** The output of the Transformer is split into scale and shift vectors, $\gamma^l(z_{<i}^l)$ and $\beta^l(z_{<i}^l)$, which parameterize an affine transformation:

$$z_i^{l+1} = \left(z_i^l - \beta^l(z_{<i}^l)\right) \cdot \exp(-\gamma^l(z_{<i}^l)).$$

   This results in a triangular Jacobian with efficient log-determinant computation.
3. **Permutation:** After each flow block, a fixed permutation reversing the order of the variables is applied to the sequence dimensions to improve expressiveness across layers.

This additive conditioning mechanism preserves causality, introduces minimal architectural overhead, and allows the model to adaptively shift the transformation of $y$ based on the input history. This structure preserves TARFLOW's benefits: exact likelihoods via tractable Jacobians, causal modelling of temporal dependencies, and a parallelisable attention-based architecture.

### C.2  TABULAR EXPERIMENT DETAILS

For the tabular benchmarks, we closely follow the setup used in CONTRA (Fang et al., 2024). Specifically, we employ a discrete normalising flow architecture (Dinh et al., 2017), which enables efficient sampling and density evaluation. Each coupling layer uses two separate neural networks to parameterise the scale and shift transformations. These networks consist of two hidden layers with 512 hidden units and ReLU activations. We use 9 coupling layers and a batch size of 512 across all datasets. Training is performed for 200 epochs using the Adam optimiser with a learning rate of

$10^{-3}$, along with an exponential learning rate scheduler that decays by a factor of 0.999 after each epoch.

Each dataset is split into 60% training, 20% calibration, and 20% test sets. For evaluating region areas in normalising flow-based methods (JAPAN, CONTRA, PCP), we generate 3,000 Monte Carlo samples per test point. We report coverage and area statistics based on 25 random seeds for the main experiments, and 5 random seeds for the various JAPAN extensions.

For baselines that require additional modelling, we use the following procedures: for **JANET**, which involves training a model on residuals, we use 60% of the calibration set for fitting a Lasso-based regressor (from `scikit-learn`) and the remaining 40% for calibration. Hyperparameters are selected using cross-validation. For **CopulaCPTS**, we similarly use 60% of the calibration set to train the copula model, and the remaining 40% for conformal calibration. For all JAPAN extensions evaluated on the Energy and RF4D datasets, we use a simple linear regressor from `scikit-learn` as the fixed base model $\mu(x)$ to generate point predictions used in the conditional and posterior variants.

### C.3    Time series experiment details

For the time series benchmarks, we use an autoregressive flow architecture adapted from TARFlow (see Section C.1). Each model is trained for 200 epochs using a batch size of 128 and a learning rate of $10^{-3}$, with an exponential learning rate scheduler (decay rate of 0.999) applied at each epoch. We first validated the hyperparameters on a simulated dataset and then reused them across all real-world experiments.

The attention-based flow model includes 10 autoregressive flow layers. Within each attention block, we use 2 attention heads, with 32-dimensional head size and 64-dimensional time-step embeddings (via expansion). Each residual block in the flow is expanded by a factor of 2 in hidden size. The input conditioning signal is processed through a context encoder comprising 2 stacked LSTM layers, each with a hidden dimension of 64.

For methods with secondary models such as **JANET**, we allocate 60% of the calibration set to train the residual regressor and reserve 40% for calibration. A Lasso-based regression model from `scikit-learn` is used, with hyperparameter tuning performed via cross-validation. Similarly, for **CopulaCPTS**, we use 60% of the calibration set to fit the copula model and the remaining 40% for calibration. We evaluate all methods across 25 random seeds for the main experiments and report mean coverage and prediction area across runs.

### C.4    Toy datasets experiment details

We evaluate our method across five synthetic 2D distributions: **Spiral**, **Checkerboard**, **Moons**, **Circles**, and **Crescent**. Each dataset consists of 10,000 samples, split into 6,000 for training, 2,000 for calibration, and 2,000 for testing. To visualise the empirical density, we also generate $n_{\text{sim}} = 1000$ samples for each dataset.

For the density model, we use a normalising flow architecture composed of 4 coupling layers, each with 2 residual blocks and hidden layer dimensionality of 32. All models are trained using the Adam optimiser with a batch size of 512 for 200 epochs. We use a learning rate of $10^{-3}$ with an exponential learning rate scheduler (decay rate of 0.999) applied at each epoch.

**Compute details:**    We conducted all experiments using NVIDIA A100 GPUs. Including failed runs, hyperparameter tuning, and iterative experimental development, the total compute time across all models is estimated to be approximately 30 GPU-days.

## D    Dataset details

### D.1    Tabular datasets

We evaluate our models on several standard tabular datasets frequently used in probabilistic regression and conformal prediction literature:

| Dataset | dim(x) | dim(y) | Type | $n_1$ (Train) | $n_2$ (Calibration) | $n_3$ (Test) |
|---|---|---|---|---|---|---|
| Energy | 8 | 2 | Tabular | 460 | 154 | 154 |
| 2D RF | 20 | 2 | Tabular | 5403 | 1801 | 1801 |
| 4D RF | 20 | 4 | Tabular | 5403 | 1801 | 1801 |
| SCM20D | 61 | 2 | Tabular | 3000 | 1000 | 1000 |
| COVID | $100 \times 1$ | $10 \times 1$ | Time Series | 200 | 100 | 80 |
| Particle-1 | $55 \times 2$ | $4 \times 2$ | Time Series | 2000 | 500 | 500 |
| Drone | $57 \times 2$ | $3 \times 2$ | Time Series | 600 | 200 | 200 |
| Pedestrian | $16 \times 2$ | $4 \times 2$ | Time Series | 891 | 200 | 200 |

Table 11: Summary of dataset structures used in our experiments. Datasets are grouped into tabular and time series types. Dimensions and splits are given for input ($x$), output ($y$), and each of the train, calibration, and test sets. For time series datasets, $a \times b$ reflects the time steps $\times$ the dimensionality of the time series.

- **Energy** (Tsanas & Xifara, 2012): This dataset consists of measurements from various buildings to predict heating and cooling loads. It includes eight features such as relative compactness, surface area, wall area, and roof area. The task is to predict the energy efficiency of buildings based on their design parameters.
- **RF2D** and **RF4D** (Spyromitros-Xioufis et al., 2016): These are multivariate regression benchmarks derived from the Mississippi River network. The dataset consists of over a year of hourly river flow observations from eight measurement sites. In the 2D RF setup, the model is trained to predict river flows 48 hours ahead at 2 downstream sites using past flows. In 4D RF, the forecasting is extended to 4 downstream sites.
- **SCM20D** (Spyromitros-Xioufis et al., 2016): This dataset comes from the 2010 Trading Agent Competition in Supply Chain Management. It contains 5,000 records, with the goal of predicting the average price of a supply chain product 20 days into the future based on a set of observed features.

### D.2 TIME SERIES DATASETS

We evaluate our method on a diverse set of synthetic and real-world time series forecasting datasets:

- **COVID** (Stankeviciute et al., 2021): We follow the experimental setup of English et al. (2024), predicting daily case counts in UK regions. The input consists of 100 days of historical data, and the model forecasts the next 10 days. The dataset includes 200 training series, 100 for calibration, and 80 for testing.
- **Particle-1** (Sun & Yu, 2024): These datasets simulate interacting particle systems using the setup of (Kipf et al., 2018). For both, the model predicts future positions $\mathbf{y}_{t+1:t+h}$ where $t = 55$ and $h = 4$. We added Gaussian noise with standard deviation $\sigma = 0.01$ to the dynamics. Each dataset includes 5,000 samples, with a 60/20/20 train/calibration/test split.
- **Drone** (Sakai et al., 2018): Simulated drone trajectories are generated using the Python-Robotics simulator. At each time step, the input is a 10-dimensional vector ($h = 57$) and the prediction target is a 3-dimensional vector ($\mathbf{y}_t \in \mathbb{R}^3$).
- **Pedestrian** (van den Berg et al., 2008): Forecasts human trajectories based on data from the ORCA simulator. Each sequence consists of 2D position measurements over $T = 20$ time steps for 1,291 pedestrians. This dataset captures multimodal future behaviour and dense agent interactions.

## E CALIBRATION PLOTS AND AREAS UNDER DIFFERENT THRESHOLDS

To assess the stability of different methods, we evaluate their calibration performance across a range of significance levels $\epsilon \in [0.05, 0.95]$. Specifically, we plot calibration curves to measure empirical coverage and visualise the corresponding prediction areas at each level, providing insight into both validity and efficiency under varying uncertainty thresholds.

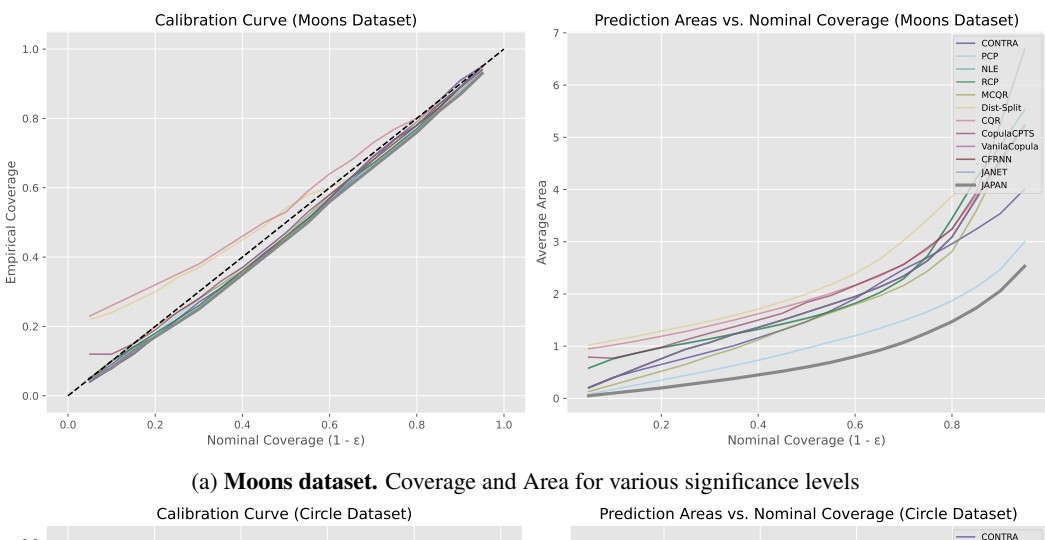

(a) **Moons dataset.** Coverage and Area for various significance levels

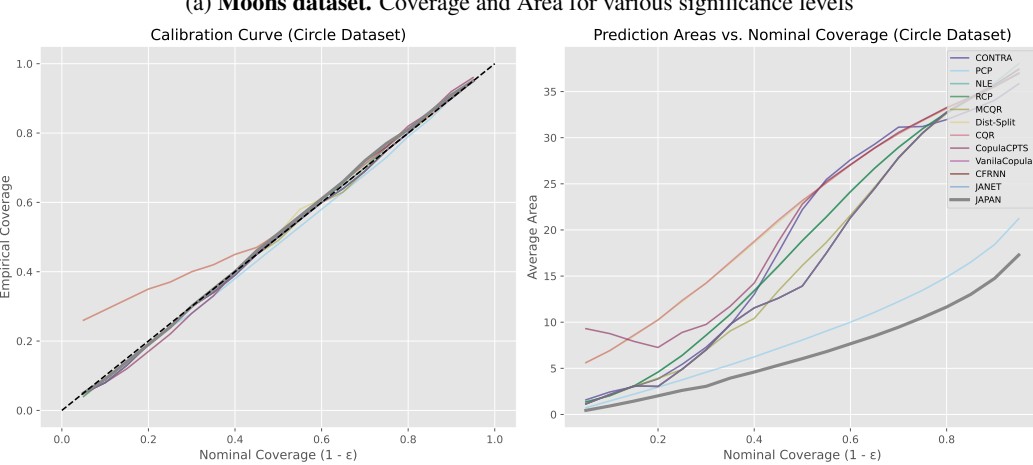

(b) **Circle dataset.** Coverage and Area for various significance levels

Figure 10: **Calibration and efficiency curves across toy datasets.** The left panels shows the calibration curve, plotting empirical versus nominal coverage for significance levels $\epsilon \in [0.05, 0.95]$. A perfectly calibrated method lies along the dashed diagonal. The bottom panel shows the corresponding average prediction region areas. JAPAN (shown with a bold line) achieves consistently valid coverage across all levels while maintaining lower or comparable region volumes relative to other methods, highlighting its balance between calibration and efficiency.

Figures 10 to 15 show these curves across multiple datasets. In each plot, the left panel presents the calibration curve, where perfect calibration corresponds to the dashed diagonal. The bottom panel reports the average prediction area across the test set at each significance level. JAPAN is highlighted with a bold grey line for easy comparison.

JAPAN consistently maintains empirical coverage close to the desired level across all thresholds, demonstrating strong calibration stability. In contrast, several baselines—including CQR, Dist-Split, and Copula-based methods—exhibit over-coverage at higher significance levels, indicating instability and excessive conservativeness. Notably, JAPAN also achieves the smallest or among the smallest prediction areas across almost all significance levels.

USE OF LARGE LANGUAGE MODELS

We used large language models (LLMs) solely for writing assistance, limited to polishing grammar and improving clarity of presentation. No LLMs were used for research ideation, theoretical results, experiments, or analysis.

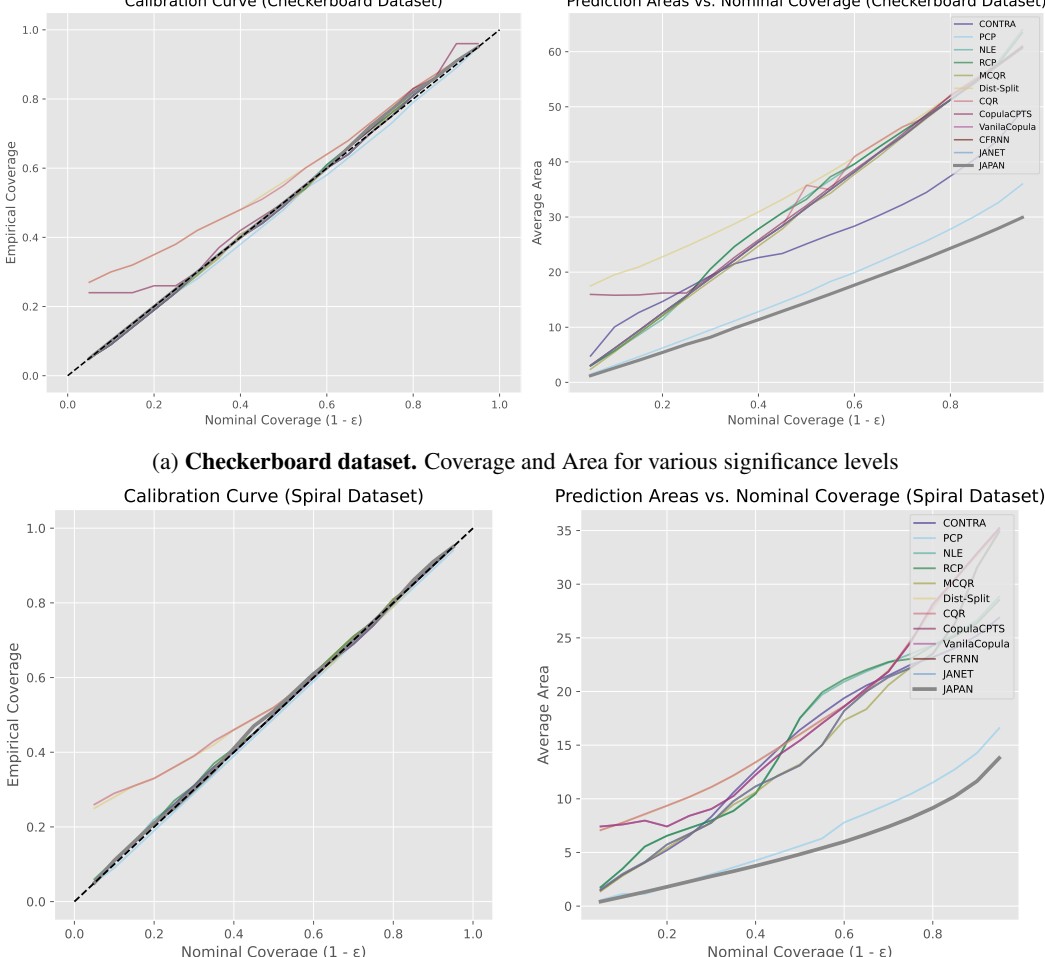

(a) **Checkerboard dataset.** Coverage and Area for various significance levels

(b) **Spiral dataset.** Coverage and Area for various significance levels

Figure 11: **Calibration and efficiency curves across toy datasets.** The left panels shows the calibration curve, plotting empirical versus nominal coverage for significance levels $\epsilon \in [0.05, 0.95]$. A perfectly calibrated method lies along the dashed diagonal. The bottom panel shows the corresponding average prediction region areas. JAPAN (shown with a bold line) achieves consistently valid coverage across all levels while maintaining lower or comparable region volumes relative to other methods, highlighting its balance between calibration and efficiency.

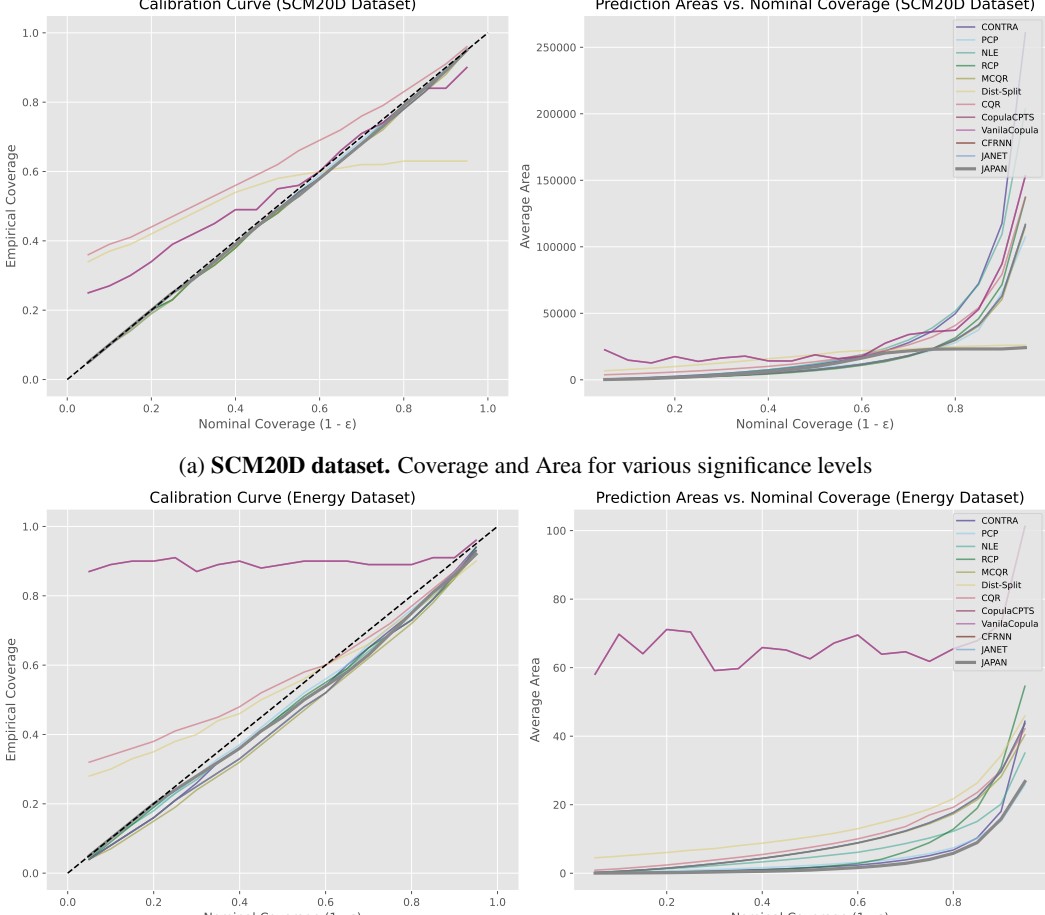

(a) **SCM20D dataset.** Coverage and Area for various significance levels

(b) **Energy dataset.** Coverage and Area for various significance levels

Figure 12: **Calibration and efficiency curves across tabular datasets.** The left panels shows the calibration curve, plotting empirical versus nominal coverage for significance levels $\epsilon \in [0.05, 0.95]$. A perfectly calibrated method lies along the dashed diagonal. The bottom panel shows the corresponding average prediction region areas. JAPAN (shown with a bold line) achieves consistently valid coverage across all levels while maintaining lower or comparable region volumes relative to other methods, highlighting its balance between calibration and efficiency.

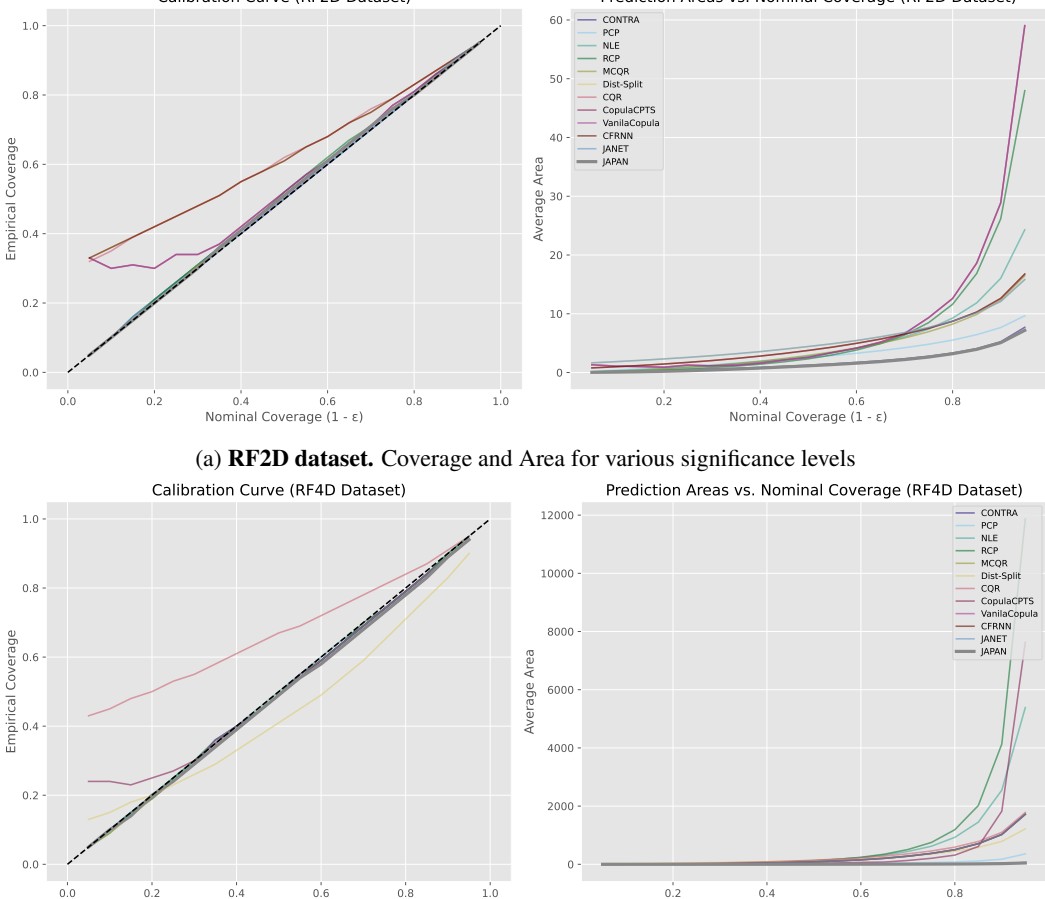

(a) **RF2D dataset.** Coverage and Area for various significance levels

(b) **RF4D dataset.** Coverage and Area for various significance levels

Figure 13: **Calibration and efficiency curves across tabular datasets.** The left panels shows the calibration curve, plotting empirical versus nominal coverage for significance levels $\epsilon \in [0.05, 0.95]$. A perfectly calibrated method lies along the dashed diagonal. The bottom panel shows the corresponding average prediction region areas. JAPAN (shown with a bold line) achieves consistently valid coverage across all levels while maintaining lower or comparable region volumes relative to other methods, highlighting its balance between calibration and efficiency.

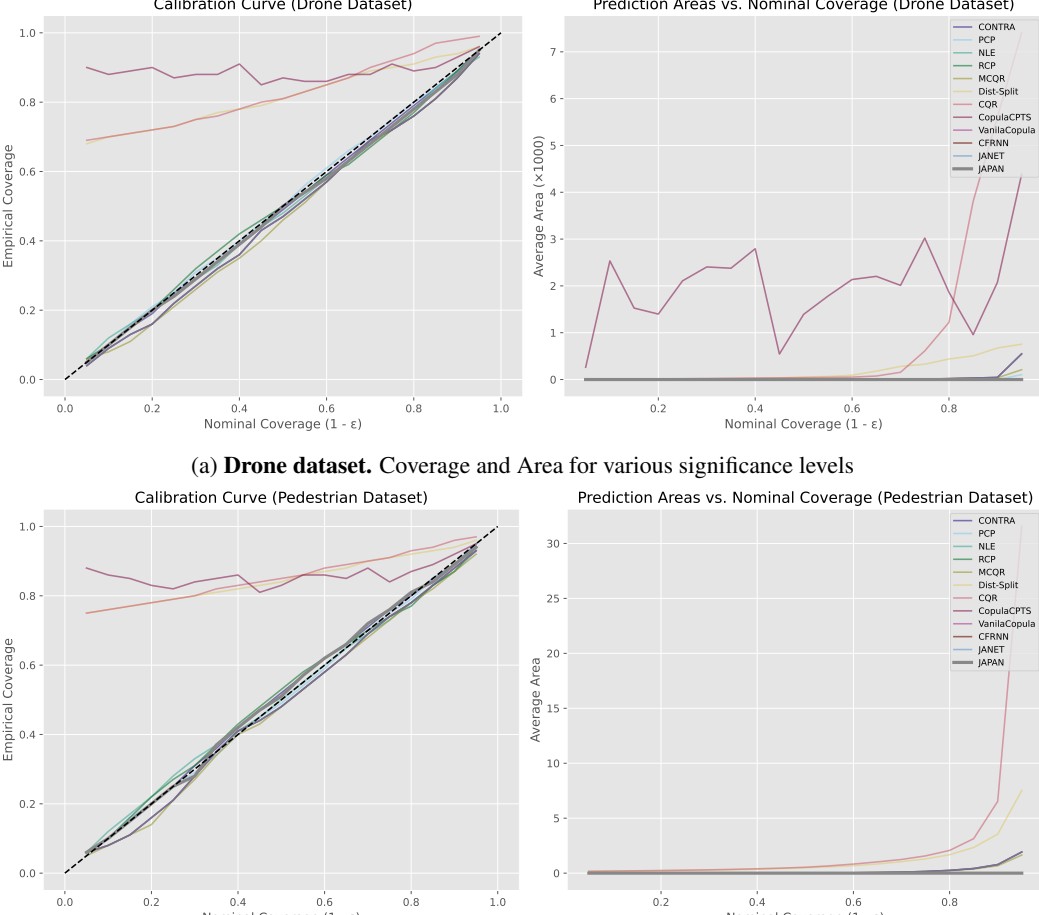

(a) **Drone dataset.** Coverage and Area for various significance levels

(b) **Pedestrian dataset.** Coverage and Area for various significance levels

Figure 14: **Calibration and efficiency curves across time series datasets.** The left panels shows the calibration curve, plotting empirical versus nominal coverage for significance levels $\epsilon \in [0.05, 0.95]$. A perfectly calibrated method lies along the dashed diagonal. The bottom panel shows the corresponding average prediction region areas. JAPAN (shown with a bold line) achieves consistently valid coverage across all levels while maintaining lower or comparable region volumes relative to other methods, highlighting its balance between calibration and efficiency.

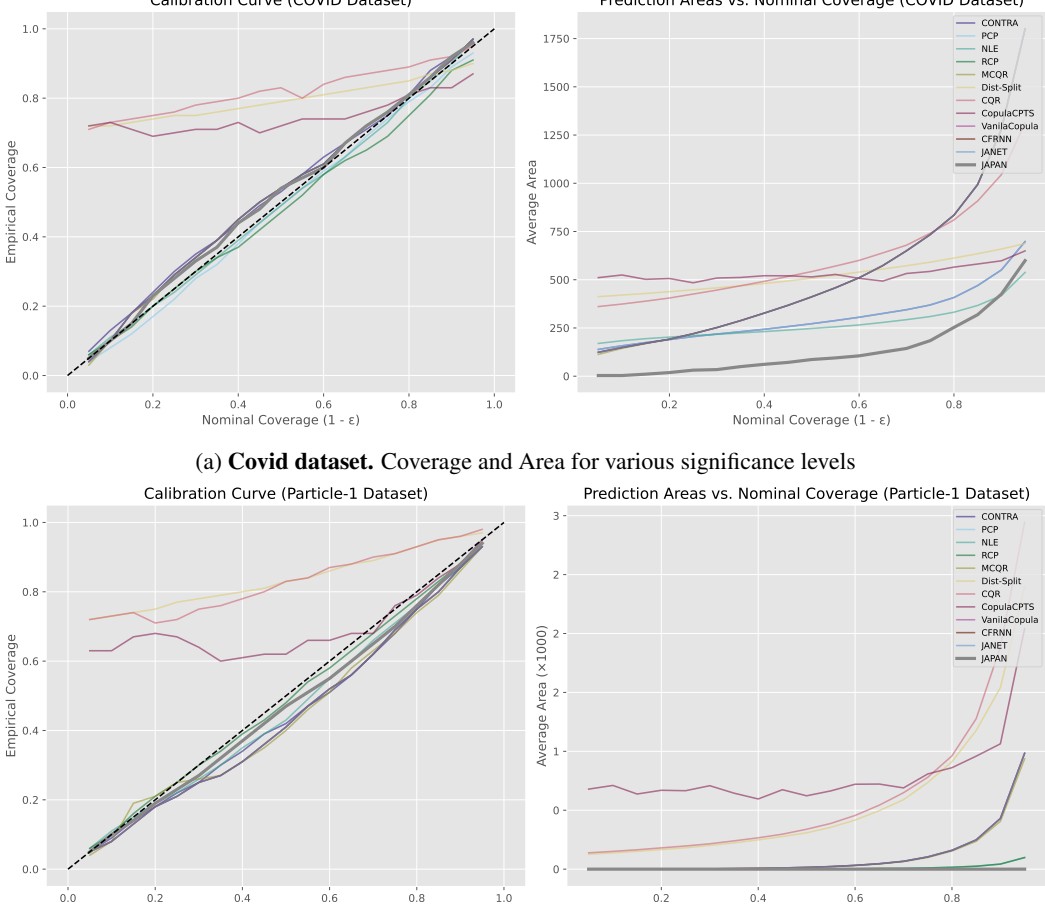

(a) **Covid dataset.** Coverage and Area for various significance levels

(b) **Particle-1 dataset.** Coverage and Area for various significance levels

Figure 15: **Calibration and efficiency curves across time series datasets.** The left panels shows the calibration curve, plotting empirical versus nominal coverage for significance levels $\epsilon \in [0.05, 0.95]$. A perfectly calibrated method lies along the dashed diagonal. The bottom panel shows the corresponding average prediction region areas. JAPAN (shown with a bold line) achieves consistently valid coverage across all levels while maintaining lower or comparable region volumes relative to other methods, highlighting its balance between calibration and efficiency.

