# OpenReview forum: "JAPAN: Joint Adaptive Prediction Areas with Normalising Flow"
_ICLR.cc/2026/Conference — ICLR 2026 Poster_

### Official Review · Reviewer_AaZM · 2025-10-30

**Soundness:** 2
**Presentation:** 2
**Contribution:** 2
**Rating:** 4
**Confidence:** 3

**Summary:**

The paper introduces a conformal prediction framework that constructs prediction sets in the output space using the log-likelihood from normalizing flows as the conformity score. This design avoids fixed geometric assumptions and enables multi-modal, disconnected prediction regions. The authors provide a rank-preserving efficiency bound and propose a latent space estimator for computing region area. Experiments on several regression and time-series datasets show smaller regions at comparable coverage.

**Strengths:**

1. This paper replaces residual based conformity scores with output space likelihoods from a normalizing flow is a simple yet powerful idea that removes restrictive geometric assumptions and enables nonconvex, multi-modal prediction regions.
2. The work subsumes prior CP variants (residual, latent, posterior) as special cases, and the paper demonstrates several practical extensions including conditional andτ-adaptive versions.
3. The experiments span both regression and time series tasks, using eight diverse datasets, and consistently show smaller prediction areas at target coverage.

**Weaknesses:**

1. The paper provides intuitive rank-preserving arguments but stops short of offering stronger formal results, there is no finite-sample or conditional coverage bound. Besides the author should incorporate more theoretical evidence.
2. The efficiency claim hinges on the flow preserving likelihood ranking. There is no evidence that this holds across architectures, capacities, or under mild misspecification. A few diagnostic plots or ablations could make the claim much more convincing.
3. Although the experiments cover many datasets, several baselines (notably RCP, MCQR, and CopulaCPTS) are numerically unstable or poorly tuned. As a result, it’s unclear whether the observed area reductions stem from the proposed idea or from implementation details.

**Questions:**

See above

---

> ### Author Response · Authors · 2025-11-19
>
> Thank you for the review. We address your queries below.
>
> **1. The paper provides intuitive rank-preserving arguments but stops short of offering stronger formal results, there is no finite-sample or conditional coverage bound. Besides the author should incorporate more theoretical evidence.**
>
> **Response:** Since JAPAN is built upon inductive conformal prediction, **finite-sample marginal coverage** follows immediately from the exchangeability of the conformity scores. This holds without any additional assumptions and is guaranteed regardless of the underlying distribution. We will make it clearer in the JAPAN.
>
> Regarding conditional coverage, we want to clarify that our work does **not** aim to provide conditional guarantees. Prior work has shown that achieving conditional coverage with conformal prediction is generally impossible without imposing strong and often unrealistic structural assumptions on the conditional distribution. This limitation applies to all CP methods.
>
> Nevertheless, we explored a direction that moves *closer* to conditional validity. Our $\tau$-adaptive extension assumes that the conditional distribution $Y \mid X=x$ changes according to approximate affine transformations in latent space. Under this assumption, **local information** about these transformations can be used to adapt the threshold. We demonstrated this behaviour on a controlled simulation example in **Section A.6, Table 7** (also added here for reference), showing that the adaptive procedure improves conditional alignment for randomly selected test instances. A comprehensive conditional-coverage theory for richer families of transformations is an interesting direction for future work.
>
> **Table 7: The conditional coverage for a toy example, where our assumptions (latent mapping and
> hence τ is the same for different values of x) are satisfied.**
> $$ \\begin{array}{|c|c|c|} \\hline \\text{Method} & \\text{Width} & \\text{Coverage} \\\\ \\hline \\text{JAPAN (x = 1)} & 3.99 \pm 0.01 & 0.90 \\\\ \\text{JAPAN (x = 0.5)} & 1.64 \pm 0.01 & 0.90 \\\\ \\hline \\end{array} $$
>
> ---
> **2.	The efficiency claim hinges on the flow-preserving likelihood ranking. There is no evidence that this holds across architectures, capacities, or under mild misspecification. A few diagnostic plots or ablations could make the claim much more convincing.**
>
> **Response:** We believe there may be a slight misunderstanding. Normalising flows, particularly discrete flows, are trained via **maximum likelihood**, so the resulting likelihood scores naturally preserve the ranking structure induced by the learned density. Even for continuous-time flow models, recent results [1, 2] show score-matching and vector-field–based training also control likelihood error, so the simply training the models is sufficient.
>
> In our experiments, we consider **three completely different flow architectures with different neural capacities**:
>
> - spline-based coupling flows for toy experiments,
> - RealNVP-style affine coupling flows for tabular datasets, and
> - our TARFlow transformer-based architecture for time-series trajectories (Section C1).
>
> These models differ significantly in capacity, architecture, and expressiveness, yet JAPAN consistently maintains strong efficiency. We also performed an ablation on **model misspecification**, where the flow was intentionally underfitted. As shown in **Table 8 and 9** (with more details in **Section A.7**) , all models degrade under this test, but JAPAN still retains substantially better region efficiency, indicating robustness to imperfect likelihood estimation. The reason is that even a partially trained flow preserves the *ordering* of likelihood values sufficiently well for JAPAN to exploit.
>
> **Table: 8 Coverage and prediction region area on the Energy and RF4D datasets. Lower area indicates better efficiency at the same coverage.**
>
> $$
> \begin{array}{lcccc}
> \textbf{Method} & \textbf{Coverage (Energy)} & \textbf{Area (Energy)} & \textbf{Coverage (RF4D)} & \textbf{Area (RF4D)} \\\\
> \hline
> \text{JAPAN}   & 0.89 \pm 0.01 & 25.61 \pm 2.44  & 0.90 \pm 0.01 & 407.68 \pm 44.19 \\\\
> \text{CONTRA}  & 0.89 \pm 0.01 & 28.13 \pm 6.18  & 0.90 \pm 0.01 & 690.99 \pm 108.35 \\\\
> \text{CPTS}    & 0.88 \pm 0.01 & 40.63 \pm 12.33 & 0.91 \pm 0.01 & 2991.77 \pm 5123.00 \\\\
> \end{array}
> $$

---

> > ### Author Response · Authors · 2025-11-19
> > **continued comment**
> >
> > **Table: 9 Underfitted modelling after decoupling scoring and predictor model: Coverage and prediction region area on the Energy and RF4D datasets. We chose two non-flow-based baselines that worked the best in this setting. Lower area indicates better efficiency at the same coverage**
> >
> > $$
> > \begin{array}{lcccc}
> > \textbf{Method} & \textbf{Coverage (Energy)} & \textbf{Area (Energy)} & \textbf{Coverage (RF4D)} & \textbf{Area (RF4D)} \times 1000 \\\\
> > \hline
> > \text{JAPAN} & 0.90 \pm 0.01 & 76.19 \pm 4.92 & 0.89 \pm 0.01 & 9.91 \pm 1.12 \\\\
> > \text{Dist-Split} & 0.89 \pm 0.03 & 151.26 \pm 56.65 & 0.83 \pm 0.01 & 20.69 \pm 3.14 \\\\
> > \text{RCP} & 0.87 \pm 0.02 & 124.46 \pm 29.45 & 0.90 \pm 0.01 & 77.15 \pm 8.72 \\\\
> > \end{array}
> > $$
> >
> >
> >
> > ---
> > **3. Although the experiments cover many datasets, several baselines (notably RCP, MCQR, and CopulaCPTS) are numerically unstable or poorly tuned. As a result, it’s unclear whether the observed area reductions stem from the proposed idea or from implementation details.**
> > **Response:** For all baselines: RCP, MCQR, CopulaCPTS, and others; we relied entirely on **official public implementations** or, when not publicly available, on **code directly provided by the authors**. We carefully followed the recommended training and evaluation settings from each paper. To avoid implementation bias, the predictive model was fixed, and **only the conformity-scoring procedures differed**, ensuring a fair and controlled comparison.
> >
> > As expected, several baselines cannot produce **disjoint prediction regions**, which is intrinsic to their modelling geometric assumptions. This is clearly visible in our toy examples **(Figure 1and Figure 4)**, where geometric restrictions in the baseline methods prevent them from adapting to multimodal or disconnected structures in the target distribution.
> >
> > Regarding numerical instability, this was **not a general issue** but occurred in specific cases. For example, RCP struggled on the COVID-19 trajectory dataset, which is inherently the most challenging among our time-series tasks and was not the intended domain of the original method (which was developed for multivariate regression). MCQR, and CopulaCPTS sometimes produced large regions in conditional models; this behaviour is expected because flow-based approaches were trained on top of other point predictors and acted as correctors of the base predictive model. Naturally, the non-flow-based baselines were not better in the conditional setting from **section A.5 or 5.1**
> >
> > Taken together, the results reflect conceptual differences between the methods rather than implementation. The observed area reductions stem from JAPAN’s density-based region construction rather than from tuning details.
> >
> > We thank you again for reviewing our work. Please let us know if we misunderstood any of your questions, or if you have any follow up on our responses, we will be happy to provide any further clarification.
> >
> > [1] Diffusion Models are Minimax Optimal Distribution Estimators, K. Oko, et al
> >
> > [2] Flow matching achieves almost minimax optimal convergence, K. Fukumizu et al

---

### Official Review · Reviewer_EZpP · 2025-10-30

**Soundness:** 3
**Presentation:** 3
**Contribution:** 2
**Rating:** 4
**Confidence:** 4

**Summary:**

The author proposes JAPAN, a multidimensional conformal prediction method that is based on Normalising flows to efficiently construct prediction regions. The author argues that current multidimensional conformal prediction methods are usually imposing restrictive geometric constraints like rectangular or spherical which often fail to capture more complex, multimodal distributions, thus leading to overly conservative sets that lose efficiency. By training a flow model, it allows for complex shape of regions which focus on the high-density areas, thus greatly improving efficiency. Besides, the author provides theoretical guarantees showing that when the density estimate is correct, the resulting prediction area is optimal. Then the author validates the performance of JAPAN by comparing it with various benchmark conformal methods.

**Strengths:**

Strength:
1. Provide theoretical guarantee showing that the region is optimal when the models correctly estimate the distribution
2. Comprehensive benchmarking in the experiment showing the efficiency and accuracy of the method

**Weaknesses:**

Weakness:
1. Lack coverage guarantees which are important in conformal prediction
2. Not comparing with other related flow-based conformal prediction method

**Questions:**

1. I notice there is other conformal prediction method for multidimensional time series ''Flow-based Conformal Prediction for Multi-dimensional Time Series'', how would you compare their method with yours?

2. It seems that the theory part is lacking the coverage guarantee. Would the author explain more in detail about this?

3. While establishing the optimality of the prediction region, the author assumes that the flow model can approximate the true density. Would the author explain more about when the assumption might approximately hold?

---

> ### Author Response · Authors · 2025-11-19
>
> Thank you for the review. We address your queries below.
>
> **1. I notice there is other conformal prediction method for multidimensional time series ''Flow-based Conformal Prediction for Multi-dimensional Time Series'', how would you compare their method with yours?**
>
> **Response:**  Thank you for pointing out this related work. To clarify, the reviewer is referring to *“Flow-based Conformal Prediction for Multi-dimensional Time Series”* by Lee, Xu, and Xie, is that correct? We reviewed the available arXiv version.
>
> There are several important differences between their setting and ours. Although both papers involve time series, our work considers **multiple independent trajectories**, where each *entire* trajectory constitutes a single data point. All trajectories are assumed to be drawn from the same underlying distribution, making the resulting conformity scores **exchangeable** by default. This is the same setting used in CopulaCPTS and CFRNN.
>
> In contrast, the cited work focuses on a **single long-running time series**, where each time index is treated as a data point. In that setting, conformity scores come from a single dependent sequence, and exchangeability does not hold. Because the two works address different problem formulations, the method in [1] is not an appropriate baseline for our setting.
>
> There are also methodological differences. JAPAN works directly with the estimated conditional density $p_\theta(y \mid x)$, and prediction regions are obtained by thresholding the log-density, enabling the possibility of **disconnected** regions when appropriate. In [1], the flow is used differently: they employ a flow-matching model to transform the distribution of residuals. Similar to CONTRA, this leads to **connected** prediction regions rather than the geometry-free regions enabled by density thresholding. Experimentally, their approach is centred around flow-matching, that is, vector-field regression-based training of flows. Nevertheless, we agree that the cited work is relevant as another flow-based approach to conformal prediction, and we will include it in the related-work section.
>
> ---
> **2.	It seems that the theory part is lacking the coverage guarantee. Would the author explain more in detail about this?**
>
> **Response:**  We believe there is a misunderstanding here regarding coverage guarantees. As mentioned previously, in both settings we study: multivariate regression and multiple time-series trajectories; the conformity scores are **exchangeable**. Under exchangeability [2], inductive conformal prediction guarantees
> $$ \Pr(y \in \Gamma_\epsilon(x)) \ge 1 - \epsilon $$
> for *any* conformity score, including **a likelihood-based** one. This is the reason we did not restate the coverage guarantee explicitly in the main text, though we will make this clearer in the revision.
>
> If we were working in the same setting as [1], where the data points arise from a *single dependent sequence* and exchangeability is violated, then a different theoretical treatment would indeed be required, involving approximate coverage or mixing assumptions. However, JAPAN does **NOT** operate in that setting, and in our setting, we obtain the coverage guarantees from the application of conformal prediction by construction.

---

> > ### Author Response · Authors · 2025-11-19
> > **continued comment**
> >
> > ---
> > **3. While establishing the optimality of the prediction region, the author assumes that the flow model can approximate the true density. Would the author explain more about when the assumption might approximately hold?**
> >
> > **Response:**  We believe there may be a slight misunderstanding regarding the assumptions in  our work. While exact density estimation would of course imply exact optimality, this is **not** required. Our results rely primarily on **rank preservation**: if the estimated density preserves the ordering induced by the true density, then the thresholded prediction region is optimal (Proposition 1). Proposition 2 shows that even when the flow model does *not* approximate the true density perfectly, **approximate** rank preservation still yields **approximate** optimality. Proposition 4 further shows that bounding the misranking error directly bounds the deviation in the prediction-region volume.
> >
> > Discrete normalising flows are trained via maximum likelihood, and continuous flows also provide likelihood-based guarantees, so in practice these models tend to capture enough of the ordering structure to make rank preservation reasonable, even when not fully trained. While we typically cannot determine whether a model perfectly approximates the true density, including simple toy examples, our experiments consistently show that JAPAN produces much tighter regions than baseline methods. In Figure 1, all conformal methods use the *same* underlying predictive model, yet JAPAN achieves significantly smaller regions due to using the density structure more effectively. The visualisation shows this gap clearly. This trend persists in higher-dimensional and real-world datasets where the true density cannot be visualised directly.
> >
> > Finally, in **section A.7 (Table 8 and Table 9)** (also added here for reference), where all models were intentionally trained under suboptimal conditions, JAPAN maintains better efficiency than the alternatives. This further illustrates that full density approximation is not necessary for JAPAN to perform well in practice.
> >
> > **Table: 8 Coverage and prediction region area on the Energy and RF4D datasets. Lower area indicates better efficiency at the same coverage.**
> >
> > $$
> > \begin{array}{lcccc}
> > \textbf{Method} & \textbf{Coverage (Energy)} & \textbf{Area (Energy)} & \textbf{Coverage (RF4D)} & \textbf{Area (RF4D)} \\\\
> > \hline
> > \text{JAPAN}   & 0.89 \pm 0.01 & 25.61 \pm 2.44  & 0.90 \pm 0.01 & 407.68 \pm 44.19 \\\\
> > \text{CONTRA}  & 0.89 \pm 0.01 & 28.13 \pm 6.18  & 0.90 \pm 0.01 & 690.99 \pm 108.35 \\\\
> > \text{CPTS}    & 0.88 \pm 0.01 & 40.63 \pm 12.33 & 0.91 \pm 0.01 & 2991.77 \pm 5123.00 \\\\
> > \end{array}
> > $$
> >
> > **Table: 9 Underfitted modelling after decoupling scoring and predictor model: Coverage and prediction region area on the Energy and RF4D datasets. We chose two non-flow-based baselines that worked the best in this setting. Lower area indicates better efficiency at the same coverage**
> >
> > $$
> > \begin{array}{lcccc}
> > \textbf{Method} & \textbf{Coverage (Energy)} & \textbf{Area (Energy)} & \textbf{Coverage (RF4D)} & \textbf{Area (RF4D)} \times 1000 \\\\
> > \hline
> > \text{JAPAN} & 0.90 \pm 0.01 & 76.19 \pm 4.92 & 0.89 \pm 0.01 & 9.91 \pm 1.12 \\\\
> > \text{Dist-Split} & 0.89 \pm 0.03 & 151.26 \pm 56.65 & 0.83 \pm 0.01 & 20.69 \pm 3.14 \\\\
> > \text{RCP} & 0.87 \pm 0.02 & 124.46 \pm 29.45 & 0.90 \pm 0.01 & 77.15 \pm 8.72 \\\\
> > \end{array}
> > $$
> >
> > We thank you again for reviewing our work. Please let us know if we misunderstood any of your questions, or if you have any follow up on our responses, we will be happy to provide any further clarification.
> >
> > [1] Flow-based Conformal Prediction for Multi-dimensional Time Series, Lee, Xu, et al
> >
> > [2] Algorithmic Learning in the Random World, V. Vovk, et al

---

### Official Review · Reviewer_LvS2 · 2025-10-31

**Soundness:** 3
**Presentation:** 3
**Contribution:** 3
**Rating:** 8
**Confidence:** 3

**Summary:**

The paper introduces JAPAN, a flow-based conformal prediction framework that uses density estimates (e.g., log p(y|x) from normalizing flows) as conformity scores to build compact, potentially disjoint prediction areas with finite-sample marginal coverage. It uses density-based conformity scores rather than residuals to avoid rigid geometries, enabling multimodal and context-adaptive regions.
The method constructs prediction areas by thresholding the estimated density and compute their volume efficiently via sampling in the flow’s latent space. It also provide theoretical motivation for efficiency under rank-preserving mappings and bounded misranking. The paper demonstrates versatility through variants: unconditional density, conditional on base predictions, posterior density, latent-space scoring, and a “tau-adaptive” variant to adjust thresholds with context. It conducts extensive experiments empirically evaluating across multivariate regression and time-series forecasting, reporting consistent calibration and smaller areas than strong baselines.

**Strengths:**

The strengths of the paper include:
1. JAPAN’s “geometry-free” density-thresholded sets can be disjoint and adapt to x, and the framework accommodates several practical scoring variants (conditional, posterior, latent), which broadens applicability.
2. Proposition 3 gives a tractable estimator via latent-space sampling and change-of-variables of the area/volume.
3. The experimental results consistently show comparable coverage and significantly smaller areas than baselines like CQR, PCP, CONTRA, copula-based methods, etc.

**Weaknesses:**

The main weakness is that the theoretical assumptions in the paper are strong and somewhat informal: Propositions 1 and 2 rely on strict monotonic transformations and density homogeneity assumptions. Proposition 4 uses a misranking mass argument without explicit finite-sample rates or dimension dependence; these read more as plausibility results than tight guarantees. Also, the claimed equivalence to CONTRA is supported empirically but not formalized.

**Questions:**

Could you include a formal argument showing JAPAN in latent is equivalent to CONTRA,
or CONTRA is a special case within JAPAN, and under what conditions?

What is the variance of the area estimator (Proposition 3) ? This estimator appears unstable due to the appearance
of the density function in the denominator.

Could you please clarify the second condition in the proof of Proposition 1 (page 24), which states that the conditional densities
p(y∣x) are homogeneous across x? Additionally, could you provide some examples where this condition is satisfied?

JAPAN appears to only provide marginal guarantee. What about conditional converage?

---

> ### Author Response · Authors · 2025-11-19
>
> Thank you for the review. We address your queries below.
>
> **1. Could you include a formal argument showing JAPAN in latent is equivalent to CONTRA, or CONTRA is a special case within JAPAN, and under what conditions?**
>
> **Response:** We thank the reviewer for this suggestion. The relationship between JAPAN and CONTRA can indeed be articulated more formally. To make the connection between CONTRA and JAPAN precise, we consider the version of JAPAN that uses the **latent Gaussian density** as the conformity score ( Latent Modelling setup in **section A.5**). In this setting, we can show that CONTRA is a special case of JAPAN because the Gaussian-density-based scores induce exactly the same ranking as the latent-space distance scores used by CONTRA.
>
> Recall that $h$ denote the normalising-flow mapping conditioned on $x$, so that
> $$
> z = h(y,x), \qquad z \sim p_Z,
> $$
> where $p_Z$ is the base distribution. We assume that $p_Z$ is an isotropic Gaussian,
> $$
> p_Z(z) = \frac{1}{(2\pi)^{d/2}} \exp\Big( -\frac{1}{2}\|z\|_2^2 \Big),
> $$
> so its log-density is
> $$
> \log p_Z(z) = -\frac{1}{2}\|z\|_2^2 - \frac{d}{2}\log(2\pi).
> $$
>
> - **CONTRA’s conformity score** (latent-ball version) can be written as
>
> $$
> \alpha_{\text{CONTRA}}(y \mid x) =  - \|h(y,x)\|_2.
> $$
>
> Note that we change the original non-conformity score to a conformity score for the comparison by changing the sign.
>
> - **JAPAN’s latent-density score** in this setting is
>
> $$
> \alpha_{\text{JAPAN}}(y \mid x) = \log p_Z(h(y,x))
> = -\frac{1}{2}\|h(y,x)\|_2^2 - \frac{d}{2}\log(2\pi).
> $$
>
> Then for any $y$ and $x$, we have
> $$
> \alpha_{\text{JAPAN}}(y \mid x) = \alpha_{\text{CONTRA}}(y \mid x) + \text{constant}.
> $$
>
> Crucially, for any two candidate outputs $y_1, y_2$ (and fixed $x$),
> $$
> \alpha_{\text{CONTRA}}(y_1 \mid x) \le \alpha_{\text{CONTRA}}(y_2 \mid x)
> \quad \Longleftrightarrow \quad
> \alpha_{\text{JAPAN}}(y_1 \mid x) \ge \alpha_{\text{JAPAN}}(y_2 \mid x).
> $$
> That is, the Euclidean norm in latent space and the latent Gaussian log-density induce **exactly the same ranking** over candidate $y$’s and equivalently the same prediction regions/areas.
>
> Thus, when JAPAN uses the latent Gaussian log-density as its conformity score, CONTRA’s latent-ball construction is retrieved as a **special case**. Importantly, the original version of JAPAN in **Algorithm 1** also includes the volume change component, which is the part that provides the flexibility to capture disjoint regions.
>
> We will add a concise version of this argument to the paper to make the relationship between JAPAN (in latent space) and CONTRA fully explicit.
>
> ---
> **2. What is the variance of the area estimator (Proposition 3) ? This estimator appears unstable due to the appearance of the density function in the denominator.**
>
> **Response:** While the estimator includes the density of the latent Gaussian in the denominator, instability does not arise in practice because regions with extremely small Gaussian density correspond to **very low-likelihood areas** in the data space and are naturally excluded by the thresholding step, or are compensated by the volume correction term. In other words, when the latent density is small, and the volume correction component is unity, the corresponding sample will almost surely lie *outside* the prediction region, and does not contribute to the volume estimate. This effectively truncates the estimator to the high-density region of the Gaussian, preventing the denominator from becoming problematic.
>
> As for the numerical stability of the estimator, our empirical studies showed no noticeable issues. With a sufficient number of samples and repeated computations of the prediction-region area, we consistently observed only negligible variation across runs, typically differences only in the decimal places. This provides further practical evidence that the estimator remains stable under repeated sampling. We will discuss it in the updated version of the manuscript. Additionally, we added a theoretical connection to Importance Sampling in **Section B.1** that shows the estimator has low variance as the NF transforms the data to noise, and the sampling weights can be analytically obtained via the change-of-variables formula.

---

> > ### Author Response · Authors · 2025-11-19
> > **continued comment**
> >
> > ---
> > **3. Could you please clarify the second condition in the proof of Proposition 1 (page 24), which states that the conditional densities p(y∣x) are homogeneous across x? Additionally, could you provide some examples where this condition is satisfied?**
> >
> > **Response:** Thanks for mentioning it. Our intention was to assume that the **shape** of the conditional densities $p(y \mid x)$ remains the same across different values of $x$, up to translation based transformations. More generally, this means that the conditional distributions share the same *spread* or covariance structure but may differ in location. This assumption is standard in settings where the conditional variability is constant but the conditional mean changes. Examples include Gaussian location families with fixed covariance, or conditional distributions that differ only by a shift parameter. We will revise the statement to make this assumption explicit and avoid ambiguity.
> >
> > To clarify the homogeneity condition used in Proposition 1: the assumption is that the conditional densities $p(y \mid x)$ share a common “shape’’ across different values of $x$. More formally, we assume that there exists a **base density** $\phi$ and a family of **translative transformations** $T_x$ such that
> > $$ p(y \mid x) = \phi(T_x(y)) \qquad \text{for all } x. $$
> >
> > Intuitively, this means that changing $x$ does not alter the intrinsic geometry of the distribution of $Y \mid X=x$; instead, it only shifts (or repositions) the distribution in the output space. The spread, modality, and overall structure of the density remain the same. Only its location changes through the map $T_x$. A good example could be:
> >
> > **Gaussian location families:**
> >   If $Y \mid X=x$ follows a Gaussian distribution with a fixed covariance $\Sigma$ but an $x$-dependent mean $\mu_x$, i.e.
> >   $$Y \mid X=x \sim \mathcal{N}(\mu_x, \Sigma),$$   then the transformation is simply $T_x(y) = y - \mu_x$, and the base density $\phi$ is the Gaussian with zero mean and covariance $\Sigma$.
> >
> > This assumption is used in Proposition 1 to argue that a single global likelihood threshold corresponds to a shape-preserving transformation across all $x$ values. We will revise the text to make this condition explicit and to include examples illustrating when it holds.
> >
> >
> > ---
> > **4. JAPAN appears to only provide marginal guarantee. What about conditional converage?**
> >
> > **Response:**  It is correct that JAPAN provides **marginal** coverage rather than conditional coverage. This is consistent with the vast majority of conformal prediction methods, as achieving conditional coverage is known to be impossible without strong structural assumptions. Our goal in this work was to obtain accurate and efficient **marginal** regions.
> >
> > Nevertheless, we also explored a direction that moves closer to conditional coverage through our **$\tau$-adaptive extension**. In this variant, we assume that the conditional distribution $Y \mid X=x$ changes via simple transformations (e.g., affine transformations) that can be captured in latent space. Under this assumption, adapting the threshold locally can yield regions with nice conditional behaviour (see **Table 7** in the paper, also added here for reference). We demonstrated this on a controlled simulation example where the structural assumptions are satisfied. A full theoretical treatment of conditional coverage for broader model families is beyond the scope of the present paper, but we view it as a promising line for future research.
> >
> > **Table 7: The conditional coverage for a toy example, where our assumptions (latent mapping and
> > hence τ is the same for different values of x) are satisfied.**
> > $$ \\begin{array}{|c|c|c|} \\hline \\text{Method} & \\text{Width} & \\text{Coverage} \\\\ \\hline \\text{JAPAN (x = 1)} & 3.99 \pm 0.01 & 0.90 \\\\ \\text{JAPAN (x = 0.5)} & 1.64 \pm 0.01 & 0.90 \\\\ \\hline \\end{array} $$
> >
> >
> >
> >
> > We thank you again for reviewing our work. Please let us know if we misunderstood any of your questions, or if you have any follow up on our responses, we will be happy to provide any further clarification.

---

### Official Review · Reviewer_MeP1 · 2025-10-31

**Soundness:** 3
**Presentation:** 2
**Contribution:** 3
**Rating:** 6
**Confidence:** 4

**Summary:**

This paper proposes JAPAN, a novel conformal prediction framework that defines flexible prediction regions using normalizing flows. Unlike CONTRA that relies on latent spaces to form conformity scores, JAPAN directly uses the loglikelihood as scores.

This enables the prediction sets to capture arbitrary, disjoint, and non-convex, non-connected shapes, while satisfying finitesample,
distribution-free marginal coverage guarantees.

The approach is evaluated and performed well on synthetic datasets, multivariate regression. The method has also been extended to handle interesting time series tasks.

**Strengths:**

Presentaiton-wise, the paper is well-structured and highly informative.

Essentially, JAPAN can provide flexible prediction sets because it is based on estimating conditional densities, and normalizing flow is a good tool to achieve this.

The application to Time series is very interesting. (I am not an expert of CP used in this area of application. I do see there are multiple papers cited. Could the authors briefly talk about how the exchangeability-backed CP theory works in times series applications? Any tips on checking the validity of marginal coverage in theory and in practice? Given a new timeseries, how do I know if JAPAN produced CP are trustworthy?)

The paper provides some theory of coverage guarantees, rank preservation, and a volume estimator that supports efficient sampling.

The method is evaluated on over mulitple datasets, including time series data, and compared with many alternative methods.
JAPAN consistently achieves lower prediction volume while maintaining coverage level.

**Weaknesses:**

The authors talked bout how to estimate the area of the proposed CP using Monte Carlo, a sensible solution. In contrast, there was a lack of mention of how to identify points y in the proposed CP, \Gamma_\epsilon(x). That is, even if one has the analytical form of the (log) probability density function, looking for all the points with density higher than a given threshold is itself a computing challenge. (A related problem is contour finding). I am myself not an expert, but I wonder what is the computing complexity of this step, and if this could quickly become prohibitively expensive when the dimensionality of outcome increases? Is it possible to take advantage of the smooth property of \hat{p} to get more efficient solutions? How this step is achieved and visualized in the current examples and how this can be efficiently achieved in higher-dimensional examples is something worth more discussion.

The paper does not specify whether JAPAN, CONTRA, and PCP use the same training setup. For
a fully controlled comparison, it would be helpful to clarify whether the same flow architecture and
training procedure were used across methods.

 Like most conformal methods, JAPAN does not guarantee conditional coverage in theory. But since JAPAN estimates conditional p(y|x) at its core,  perhaps some empirical studies can be added to show if it does reasonably well in conditional coverage?

 The method requires training normalizing flows and sampling extensively to approximate the region.
Runtime and efficiency were not clearly discussed in the main text.


Presentation-wise, the paper lacks consistency and rigoricity in symbol usage and mathematical languages. Most are minor issues, but worth a good round of proofreading. I am only giving three examples below, among other similar problems:

In the statement of Proposition 3, I think “z” was abused to denote both a sample of size N (z_1,\cdots, z_n) and just one instance/ in h^{-1}(z, x)? Also, both z and z_i appeared in the Area estimation formula. And mathematically, it’s not tight to write z \sim p_X(z), with the same symbol z on both sides…

The introduction, followed by a review of NF and inductive conformal prediction is generally well-done. The detailed description provided for ICP is actually for a most popular case of ICP, the split conformal. So that part could be a little misleading.

Sec 3.1 \hat{p} was not officially defined before it appears inside the conformal prediction set.

**Questions:**

My questions are embedded in my comments for Strengths and Weaknesses above. Thanks.

---

> ### Author Response · Authors · 2025-11-19
>
> Thank you for the review. We are glad you liked the presentation of our work. We address your queries below.
>
> **1. Could the authors briefly talk about how the exchangeability-backed CP theory works in times series applications?**
>
> **Response:** We would first like to clarify a key misunderstanding regarding the time-series setting in our work. In the conformal prediction literature, there are two fundamentally different ways in which time series are treated. In the first setting, we only have access to a *single* long time series, where each time step is regarded as one data point. In this case, the conformity scores are computed from successive points of the same sequence and therefore exhibit sequential dependence. Because of this serial dependence, the classical exchangeability assumption is violated, and the usual split-CP guarantees do not apply directly. Methods developed for this setting typically rely on additional assumptions such as strong mixing (which yield approximate coverage), or they attempt to maintain empirical coverage by adaptively adjusting the target significance level, either heuristically or through a learnable mechanism.
>
> In contrast, the second setting assumes access to multiple independent time-series trajectories, where each entire trajectory is treated as one data point. All trajectories are assumed to be sampled from the same underlying distribution, meaning that the resulting conformity scores are **exchangeable** and do not suffer from serial dependence. This is exactly the setting we consider in our work. Because the unit of exchangeability is the trajectory (not individual time steps), standard inductive conformal prediction applies without modification. For example, with a dataset of drone trajectories, each full trajectory is an independent draw, and conformal prediction intervals for a new trajectory satisfy the usual guarantee:
> $$ \Pr(y \in \Gamma_\epsilon(x)) \ge 1 - \epsilon. $$
>
> To summarise, our method operates in the multiple-trajectory setting, where exchangeability holds and classical conformal prediction guarantees apply as-is. Methods designed for the single-sequence setting solve a fundamentally different problem and address the lack of exchangeability, which is not relevant in our case.
>
> ---
> **2. Any tips on checking the validity of marginal coverage in theory and in practice? Given a new time series, how do I know if JAPAN produced CP are trustworthy?)**
>
> **Response:**  Regarding the validity of marginal coverage in theory and in practice, we emphasise that in the two settings we consider—multivariate regression and the multiple–time-series setting—marginal coverage holds by construction. This follows from the fact that the only assumption required by inductive conformal prediction is **exchangeability**, which is even weaker than the standard i.i.d.\ assumption commonly used in machine learning. Because each full trajectory (not each time step) is treated as one data point, and all trajectories are sampled from the same underlying distribution, exchangeability holds by construction.
>
> For a new time-series trajectory, we have already (i) trained the predictive model, (ii) calibrated the conformity scores using an exchangeable calibration set, and (iii) evaluated performance on a separate test set. Under these conditions, the resulting regions are trustworthy in exactly the same sense as any other conformal prediction method. Intuitively, conformal prediction is just searching for regions that cover a desired fraction of *unseen* data; as long as the distribution of calibration trajectories matches that of the test trajectories, the nominal marginal coverage follow under the assumption.

---

> ### Author Response · Authors · 2025-11-19
> **Continued comment**
>
> **3. In contrast, there was a lack of mention of how to identify points y in the proposed CP......efficiently achieved in higher-dimensional examples is something worth more discussion.**
>
> **Response:** In our work, we only identify points that exceed the learned density threshold for the purpose of computing or visualising the region. While a naive Monte Carlo approach would require defining a dense grid over the entire label space—which becomes intractable as dimensionality grows—we do **not** use such a grid-based strategy. Instead, our estimator samples from the *latent* Gaussian distribution used by the normalising flow.
>
> Intuitively, in a generative model, we sample from the learned data distribution, which means high-density regions are sampled more and low-density areas are sampled less. When working with flow-based models, the change-of-variable formula ensures we can sample from the learned data distribution by sampling from the noise distribution. Our area/volume estimator in **Proposition 3** performs the same function as sampling from the noise distribution is corrected by the volume correction part from the change-of-variable formula. Intuitively, this is what provides efficiency to our volume estimator.
>
> At same time, sampling also naturally concentrates on high-density areas in latent space. Because normalising flows training map high density Gaussian mass toward regions of high likelihood in the data space (as encouraged by the likelihood objective), this itself suggests more efficient sampling as low density areas are less likely to be sampled in the latent space itself and so in the data space and vice versa. In **Table 10** (we also show it here for reference) we already demonstrated that our estimator is substantially more efficient than naive sampling. Particularly, the naive grid-based sampling takes for 8-dimensional output is 1,000,000 points and requiring 32 minutes, whereas utilising the latent space only required 2.5 minutes and only 3,000 samples.
>
> \begin{array}{lcccc} \textbf{Method} & \textbf{Samples (d=4)} & \textbf{Time (minute)} & \textbf{Samples (d=8)} & \textbf{Time (minutes)} \\\\ \hline \text{Ours} & 1,000 & 0.4 & 3,000 & 2 \\\\ \text{Naive-MC} & 100,000 & 5 & 1,000,000 & 32 \\\\ \end{array}
> **Table 10:  Samples required for comparable accuracy on the test set with sampling schemes sampling,
> d = 4 and d = 8.**
>
>
>
> In our experiments we consider output dimensions up to 10, which already covers a large range of practical scenarios; most applications requiring joint prediction regions do not operate in extremely high-dimensional output spaces. Finally, when dimensionality becomes very high—as in image outputs—the **manifold hypothesis** becomes relevant: the intrinsic dimensionality of the data is typically much lower than its ambient dimension. Because our estimator samples in latent space, Proposition 3 applies to the intrinsic manifold structure, meaning that even in such high-dimensional cases the estimator remains efficient without traversing the full ambient space.
>
> ---
> **4. Like most conformal methods, JAPAN does not guarantee conditional coverage in theory. But since JAPAN estimates conditional p(y|x) at its core, perhaps some empirical studies can be added to show if it does reasonably well in conditional coverage?**
>
> **Response:** It is correct that JAPAN, like most conformal methods, does not guarantee conditional coverage in theory. Although JAPAN estimates conditional densities $p(y \mid x)$, the calibration step in Algorithm 1 is performed *marginally* over all values of $x$. This is analogous to classical regression-based conformal prediction, where the (non)conformity score is often the absolute residual $|y - f(x)|$, which also depends on $x$ but still yields only marginal coverage when calibrated globally.
>
> Therefore, we do not expect Algorithm 1 to provide conditional coverage guarantees. Nevertheless, we explored a variant with **adaptive thresholds** (Algorithm 6 and Section A.6) designed to move closer to conditional coverage, and we reported results in **Table 7** (also added here for reference). The intuition behind this adaptation is that the conditional distributions $Y \mid X=x$ often vary in ways that can be captured through approximate affine transformations in latent space. In simple simulation examples we verified that this property holds, and that adaptive thresholding indeed brings the coverage closer to a conditional one.
>
> **Table 7: The conditional coverage for a toy example, where our assumptions (latent mapping and
> hence τ is the same for different values of x) are satisfied.**
> $$ \\begin{array}{|c|c|c|} \\hline \\text{Method} & \\text{Width} & \\text{Coverage} \\\\ \\hline \\text{JAPAN (x = 1)} & 3.99 \pm 0.01 & 0.90 \\\\ \\text{JAPAN (x = 0.5)} & 1.64 \pm 0.01 & 0.90 \\\\ \\hline \\end{array} $$
>
> ---

---

> > ### Comment · Reviewer_MeP1 · 2025-11-26
> >
> > The authors clarified a few things per my questions, which I appreciate.
> >
> > I agree in practice, sampling from the Gaussian in the latent space before transforming is a good idea. This is not that important, but I'd like to mention that I don't agree with and don't understand the bracket in the sentence "normalising flows map high density Gaussian mass toward regions of high likelihood in the data space (as encouraged by the likelihood objective)" --  there is really no guarantee, especially in multi-dimensional cases. Whether this correspondence is approximately true highly depends on the (volatility of the) gradient of the normalizing flow transformation. It's more like the authors HOPE this correspondence is true, so that their sampling scheme is efficient.
> >
> > Overall, the original score of 6 still reflects my assessment of the paper.

---

> ### Author Response · Authors · 2025-11-19
> **continued comment 2**
>
> **5. Runtime and efficiency were not clearly discussed in the main text.**
>
> **Response:** Thanks for the point. We will discuss it properly in the main text. To give a brief answer here:
>
> Firstly, please note that NF was trained as a fixed base predictor for all the baslines, which makes the training cost the same with the only difference being how the conformal methods operate and if they have any additonal cost of implementation.
>
> All of our NF models were trained in under 2 minutes. While hyper parameter tuning might still be needed, it is central to any neural network-based method. Additionally. two baselines in our paper (CONTRA and PCP) also use NFs as part of their pipeline, meaning the overall cost comparable. Additionally, methods such as JANET and copula methods require training separate models for computing their scores. JANET, for example, can use linear regression, but good results are obtained using methods such as random forests, gradient boosting machine, which adds to the inference costs (on top of training a a base predictor). Also, they require splitting the calibration into two, which creates data inefficiency at the same time as a downside. So while these methods may appear simpler in model design, they do not eliminate computational costs and inference costs altogether, so the analysis is much more nuanced.
>
> On our inference: JAPAN does involve density evaluation and sampling, but thanks to batch operations, computing a prediction region for one sample involves only a single forward pass (for a batch of a thousand). For many applications, such as time series, we work on one trajectory at a time, so the inference overhead is negligible (microseconds to a second). The additional Monte Carlo integration step introduces mild overhead, but it is not prohibitive. As with many modelling decisions, there is a trade-off between computational effort and region sharpness. For example, training a neural network generally requires more compute than fitting a linear classifier, but offers better representation power. Similarly, flows allow for more adaptive and sharper uncertainty quantification.
>
> ---
> **6. the paper lacks consistency and rigoricity in symbol usage and mathematical languages. Most are minor issues, but worth a good round of proofreading. I am only giving three examples below, among other similar problems**
>
> **Response:** We thank the reviewer for pointing out inconsistencies in notation. Because the paper brings together several components: normalising flows (including continuous-time variants), conformal prediction, multivariate regression, multi-time-series modelling, and a new architecture adaptation of TARFLOW; there were a few instances of notational abuse. We will perform a careful proofreading to ensure consistency in symbols and mathematical language throughout the paper. We will also add a short subsection clarifying any remaining points where a nonstandard or overloaded notation is used.
>
> ---
> **7. The introduction, followed by a review of NF and inductive conformal prediction is generally well-done. The detailed description provided for ICP is actually for a most popular case of ICP, the split conformal. So that part could be a little misleading.**
>
> **Response:**  We kindly ask the reviewer to clarify this comment. Our understanding is that inductive conformal prediction (ICP) [1] and what is commonly called “split conformal prediction” are synonymous terms, referring to the same procedure. The section in question was intended to describe this standard ICP/split-conformal setup. If the reviewer meant something more specific, we would be happy to revise accordingly.
>
> ---
> **8. The paper does not specify whether JAPAN, CONTRA, and PCP use the same training setup. For a fully controlled comparison, it would be helpful to clarify whether the same flow architecture and training procedure were used across methods.**
>
> **Response:**  We want to clarify that all methods including JAPAN, CONTRA, PCP, use **identical training setups**. More precisely, the normalising-flow model is trained once, and all baselines compute their prediction regions using this same trained model. This ensures that the comparison is completely controlled and isolates differences arising solely from the conformal procedures rather than from model training.
>
> We thank you again for reviewing our work. Please let us know if we misunderstood any of your questions, or if you have any follow up on our responses, we will be happy to provide any further clarification.
>
> [1] Algorithmic Learning in the Random World, V. Vovk, et al

---

> ### Author Response · Authors · 2025-11-27
>
> Thanks for your response. Please let us clarify what we meant by the statement: “normalising flows map high-density Gaussian mass toward regions of high likelihood in the data space (as encouraged by the likelihood objective).”
>
> We were referring to the **training objective** of normalising flows. The maximum-likelihood objective optimises the log-density
> $$
> \log p(y|x) = \log p_Z(h(y,x)) + \log\ \big|\det J_{h}(y,x)\big|,
> $$
> where $z = h(y,x)$ is the latent mapping.
>
> Conversely, the negative log-likelihood objective for normalising flows minimises the two components simultaneously, i.e., the norm of the transformed noise variable $z$ together with the volume-change term. The first term, which corresponds to reducing the norm of the transformed data, encourages the transformed variables to follow the Gaussian prior. Whereas the second term, which corresponds to the volume change, encourages the flow to learn a transformation that pushes higher-likelihood regions of the data distribution toward regions of higher Gaussian density in latent space, and vice versa. This is the interpretation we intended with our original statement. Moreover, works such as [1,2] have shown forms of likelihood control and minimax convergence for continuous flow models (score-matching and velocity-regression loss functions), which shows why this tends to happen in these models.
>
> That said, we agree that this correspondence does not hold perfectly everywhere. In particular, there are cases where low-density regions in the data distribution must be mapped to higher-density regions in latent space, depending on the structure of the flow transformation. This is especially visible in certain toy examples, such as the spiral dataset, where the optimal transformation necessarily sends large low-density parts of the spiral into regions of higher Gaussian density. In such situations, high-density latent regions are sampled more frequently; however, they still correspond to large portions of the data space, as the total probability mass must be conserved. Sampling these regions is, in fact, necessary; without doing so, we would risk including those low-density parts in our computed regions (if they are below the threshold), leading to larger areas and failing to provide the **disjoint property** that we aim to achieve in our method.
>
> Please note that if such regions exist, the large volume of the data space would occupy only an infinitesimal volume in the latent space. In this case, we are not sampling those regions more than necessary, thereby retaining efficiency. This finding is also consistent with our empirical observations.
>
> [1] Diffusion Models are Minimax Optimal Distribution Estimators, K. Oko, et al
>
> [2] Flow matching achieves almost minimax optimal convergence, K. Fukumizu et al

---

> ### Author Response · Authors · 2025-11-27
> **Alternative view of area/volume estimator**
>
> We thank the reviewer again for their follow-up to our earlier response. Please let us offer an alternative perspective on the estimator's efficiency.
>
> Whilst sampling from the latent space is effectively equivalent to sampling from the data distribution, and contributes to the efficiency of the estimation process. We can also view the volume/area estimator as an **importance sampling estimator**, which helps explain both its correctness and practical efficiency.
>
> Using the change-of-variables formula, the volume of the prediction region
>
> $$
> \Gamma_\epsilon(x) = \\{ y : \log p_\theta(y \mid x) \ge \tau \\}
> $$
>
> can be written as
>
> $$
> \text{Area}(\Gamma_\epsilon(x))
> = \int_{h(\Gamma_\epsilon(x))}
> |\det J_{h^{-1}}(z,x)|\ dz.
> $$
>
> Here, $h^{-1}$ is the transformation from the latent space to the data space, and $J_h^{-1}(z,x)$ is the Jacobian of the transformation.
>
> By introducing the latent Gaussian $p_Z(z)$ and multiplying and dividing by $p_Z(z)$, this becomes the expectation
>
> $$
> \text{Area}(\Gamma_\epsilon(x)) =
> E_{z\sim p_Z}
> \left[
> \frac{|\det J_h^{-1}(z,x)|}{p_Z(z)}
> \ \mathbf{1}(h^{-1}(z_i,x) \in \Gamma_\epsilon(x))
> \right\],
> $$
>
> which is exactly the form of an **importance sampling estimator** with proposal distribution $p_Z$. The weights
>
> $$
> w(z,x)=\frac{|\det J_h^{-1}(z,x)|}{p_Z(z)}
> $$
>
> are the corresponding importance weights, and the indicator selects points in the prediction region. Our Monte Carlo estimator is therefore an Importance Sampling estimator:
>
> $$
> \widehat{\text{Area}}(\Gamma_\epsilon(x))=
> \frac{1}{M}\sum_{i=1}^M
> w(z_i,x)\
> \mathbf{1}\ \big(h^{-1}(z_i,x)\in\Gamma_\epsilon(x)\big),
> \qquad z_i\sim p_Z.
> $$
>
> This Importance Sampling viewpoint also explains the **efficiency** of the estimator. Because normalising flows maps the data distribution to *Gaussian latent* $p_Z$, and the data variable $y$ can be expressed as a function of $z$, the Importance Sampling viewpoint of using the Gaussian as the proposal distribution. This avoids the inefficiency of naive grid-sampling or uniform sampling in the output space. In other words, the proposal distribution $p_Z$ is well-aligned with the transformation-induced distribution of the data via flows, which also reduces the variance of the estimator, as the weights are naturally obtained from the volume change component that can be analytically calculated via flows.
>
> We will add a short explanation of this importance-sampling perspective to the final version of the paper, as it cleanly justifies both the estimator’s validity and its computational efficiency.

---

### Author Response · Authors · 2025-11-27

Dear reviewers,

We thank you again for your time and effort in reviewing our work. We hope we have addressed all of your concerns. If you have further questions after our responses, please let us know. If we have addressed all of your concerns, could you please consider updating your score?

Thanks,

---

### Author Response · Authors · 2025-12-03
**Summary comment**

Dear Area Chair,

Thank you for the time and effort you put into evaluating our work. We want to summarise the main points of discussion with the reviewers in this comment.

Overall, after submitting our rebuttals, we only heard back from one of the four reviewers before the discussion period was cut short. Even with reviewer MPE1, who responded, we were only able to submit our response with a theoretical analysis, but were unable to obtain their response due to the curtailed discussion period. The main points of the discussions were:

1. **Clarity on the experimental setting:** We worked on two settings. One was a multivariate regression where the response variable $y \in \mathbb{R}^{d_y}$ conditoned on $x \in \mathbb{R}^{d_x}$. The other one is working on trajectories, simply (mentioning univariate trajectories here for notational simplicity), where $d_x$ is the dimensionality of the covariate that would refer to $1:T$ steps of $i^{th}$ trajectory and $d_y$ is the dimensionality of the response that would refer to $T+1:T+H$ steps of $i^{th}$ trajectory. Each of these trajectories is exchangeable, and each trajectory is a data point/sample, as opposed to a value at each time index $t$, which would violate the exchangeability assumption, and existing coverage guarantees for conformal prediction would not apply. We stress again, multivariate regression and working on multiple trajectories preserve exchangeability requirements. The multiple-trajectory setup is quite common in conformal literature [3,4]. We clarified it in the paper.

2.  **Efficiency of volume estimator and its variance**: With generative modelling, we sample from the learned data distribution, which means high-density regions are sampled more and low-density areas are sampled less. When working with flow-based models, the change-of-variable formula ensures we can sample from the learned data distribution by sampling from the noise distribution. Our volume estimator in **Proposition 3** performs the same function as sampling from the noise distribution is corrected by the volume correction part from the change-of-variable formula. Intuitively, this is what provides efficiency to our volume estimator. This efficiency was empirically shown in **Table 10** in the paper and also in the response to the reviewer here. Additionally, our volume estimator has a link to importance sampling, which we have added to the paper in **Section B.1** and also in our response to the reviewer, to show why our volume estimator is **efficient and has low variance**. We also added explanations on how flow-based models try to map high-density Gaussian areas to high-density regions in data distribution, which also contributes to efficiency, and an explanation of what happens when the optimal transformation can not do so. Briefly, in such cases, the infinitesimal area of Gaussian maps to a large chunk of area in the data distribution cos the total probability mass is the same and we end up exploring more area, which is needed as we seek to find the region above a certain threshold.

3. **Conditional coverage**: Aiming for conditional coverage was never the aim of this work, but instead developing a method that allows for finding disjoint regions without any assumption, a feature lacking in the existing methods (**Table 1 and Figure 1 in the main paper**). Nonetheless, we demonstrated how conditional coverage can be achieved by creating the conformal threshold as a function of the test input. We demonstrated that when the dependence on the conditional distribution is an affine function of the test input, conditional coverage can be achieved. We showed it in **Table 7** for a simple example. Note that it is already known that conformal methods can not achieve conditional coverage without structural assumptions [5].

4. **Formal proof on how CONTRA is a special version of JAPAN**: CONTRA used distances of latent mappings which followed a Gaussian distribution, whereas JAPAN (when the volume change component is zero or is not included as in **Algorithm 5/section A.5**) assigns the same ranking to latent density as CONTRA, thanks to the distribution being a standard Gaussian. We formalised that intuition and added to the paper in **Section A.5**.

5. **Clarification of the second condition in the proof of Proposition 1**: The assumption was that the shape of the distribution condition on the test input does not change, and it is the same up to a translation factor. We clarified the assumption in the paper further.

6. **Comparison with [1]**: The work is fundamentally very different from ours. We work in an entirely different setting. As mentioned in the first point of the summary, we worked with multiple trajectories, where each trajectory represents a distinct data point. [1] focuses on a single time series, where each time step is a data point, which violates exchangeability. Nonetheless, we discussed this in the related works.

**[continued in the next comment]**

---

> ### Author Response · Authors · 2025-12-03
> **Summary comment continued**
>
> 7. **Lacking coverage guarantee**: This is NOT true! We do have coverage guarantees through the use of conformal prediction. Any conformity score can provide coverage guarantees as long as the conformity scores are exchangeable [2]. Reviewer EZpP appears to have misinterpreted our time series trajectory setting as one where each time index corresponds to a data point, which violates exchangeability and necessitates new coverage guarantees, as apparent from their question on comparison with [1]. We clarified it in the paper.
>
> 8. **Using different architectures, capacity of neural networks**: We have indeed used different architectures and networks of different capacities in our work, namely-
> - spline-based coupling flows for toy experiments,
> - RealNVP-style affine coupling flows for tabular datasets, and
> - our TARFlow transformer-based architecture for time-series trajectories (Section C1).
>
> 9. **Ablation with model mis-specification**: Firstly, we point out that a conformal method calibrates for a model mis-specification. When a flow-based model is poorly trained as a predictor, the performance of all the baselines will deteriorate, including ours. We showed this result in **Table 8**, our method, JAPAN, still performed the best. Note that this setting combined the conformity scoring with the predictor as a flow-based model explicitly provides density estimates. We also performed an ablation that **decoupled the scoring rule** from the base predictor. This meant we trained a flow conditioned on the prediction of another model. We found a flow-based model still performed better, cos the additional layer of flow corrected for the prediction errors of the base model. These results can be seen in **Table 9**. The entire discussion can be seen in **section A.7** of the paper.
>
> 10. **Implementation details of other baselines**: For all baselines: RCP, MCQR, CopulaCPTS, and others; we relied entirely on official public implementations or, when not publicly available, on code directly provided by the authors. We clarified it in the paper.
>
> We addressed all these in our responses and also **updated the paper** to reflect these changes along with other minor details such as **notational clarity, and clarity on identical training setups**. For an easy overview, we add the Tables we mentioned here in the comment as well.
>
> [1] Flow-based Conformal Prediction for Multi-dimensional Time Series, Lee, Xu, et al
>
> [2] Algorithmic Learning in the Random World, V. Vovk, et al
>
> [3] Copula Conformal Prediction for Multi-step Time Series Forecasting by S Sun, et al
>
> [4] Conformal Time-Series Forecasting by K Stankeviciute, et al
>
> [5] The limits of distribution-free conditional predictive inference. R.F Barber, et al
>
> **Table 7: The conditional coverage for a toy example, where our assumptions (latent mapping and
> hence τ is the same for different values of x) are satisfied.**
> $$ \\begin{array}{lccc} \\textbf{Method} & \\textbf{Width} & \\textbf{Coverage} \\\\ \\hline \\text{JAPAN (x = 1)} & 3.99 \pm 0.01 & 0.90 \\\\ \\text{JAPAN (x = 0.5)} & 1.64 \pm 0.01 & 0.90 \\\\ \\hline \\end{array} $$
>
> **Table: 8 Coverage and prediction region area on the Energy and RF4D datasets for poorly-trained flow model. Lower area indicates better efficiency at the same coverage. JAPAN yields the lowest area.**
>
> $$
> \begin{array}{lcccc}
> \textbf{Method} & \textbf{Coverage (Energy)} & \textbf{Area (Energy)} & \textbf{Coverage (RF4D)} & \textbf{Area (RF4D)} \\\\
> \hline
> \textbf{JAPAN}   & \textbf{0.89} \pm 0.01 & \textbf{25.61} \pm 2.44  & \textbf{0.90} \pm 0.01 & \textbf{407.68} \pm 44.19 \\\\
> \text{CONTRA}  & 0.89 \pm 0.01 & 28.13 \pm 6.18  & 0.90 \pm 0.01 & 690.99 \pm 108.35 \\\\
> \text{CPTS}    & 0.88 \pm 0.01 & 40.63 \pm 12.33 & 0.91 \pm 0.01 & 2991.77 \pm 5123.00 \\\\
> \end{array}
> $$
>
> **Table: 9 Underfitted modelling after decoupling scoring and predictor model: Coverage and prediction region area on the Energy and RF4D datasets. We chose two non-flow-based baselines that worked the best in this setting. Lower area indicates better efficiency at the same coverage**
>
> $$
> \begin{array}{lcccc}
> \textbf{Method} & \textbf{Coverage (Energy)} & \textbf{Area (Energy)} & \textbf{Coverage (RF4D)} & \textbf{Area (RF4D)} \times 1000 \\\\
> \hline
> \textbf{JAPAN} & \textbf{0.90} \pm 0.01 & \textbf{76.19} \pm 4.92 & \textbf{0.89} \pm 0.01 & \textbf{9.91} \pm 1.12 \\\\
> \text{Dist-Split} & 0.89 \pm 0.03 & 151.26 \pm 56.65 & 0.83 \pm 0.01 & 20.69 \pm 3.14 \\\\
> \text{RCP} & 0.87 \pm 0.02 & 124.46 \pm 29.45 & 0.90 \pm 0.01 & 77.15 \pm 8.72 \\\\
> \end{array}
> $$
>
> **Table 10:  Samples required for comparable accuracy on the test set with sampling schemes sampling,
> d = 4 and d = 8.**
> \begin{array}{lcccc} \textbf{Method} & \textbf{Samples (d=4)} & \textbf{Time (minute)} & \textbf{Samples (d=8)} & \textbf{Time (minutes)} \\\\ \hline \textbf{Ours} & \textbf{1,000} & \textbf{0.4} & \textbf{3,000} & \textbf{2} \\\\ \text{Naive-MC} & 100,000 & 5 & 1,000,000 & 32 \\\\ \end{array}

---

### Meta-Review · Area_Chair_EMRi · 2026-01-07

**Summary:**

The following key concerns were raised by reviewers:
1. Apparent lack of theoretical coverage guarantees (AaZM, EZpP)
2. Lack of comparison to related flow-based conformal prediction methods (EZpP)
3. Insufficient evidence of likelihood ranking preservation across architectures, capacities, and model misspecification (AaZM)
4. Theoretical assumptions are strong and somewhat informal (LvS2)
5. Unclear evidence that CONTRA is a special case of the proposed method (LvS2)
6. Lack of clarity about how the prediction region can be computed (MeP1)
7. Lack of clarity about whether baselines use the same training setup (MeP1)

**Reviewer Concerns:**

The authors addressed the concerns as follows.
1. The authors clarified that this concern stems from a misunderstanding regarding the nature of coverage guarantees. Due to the exchangeability of trajectories, the proposed method is indeed equipped with a coverage guarantee.
2. The authors clarified that related work focuses on a single long-running time series rather than multiple independent trajectories as considered here. They agreed to update the related work accordingly.
3. The authors responded that normalizing flows naturally preserve likelihood scores due to their maximum likelihood training objective. Additionally, the authors provided evidence that the proposed method is effective across architectures and also an ablation on model misspecification.
4. The authors provided a detailed justification for theoretical assumptions and agreed to revise the text to make the theoretical conditions explicit. Importantly, propositions 1 and 2 do not affect the validity of the method. Instead, they show that if the flow is a good model of the true conditional density, then the area of the prediction region will be nearly optimal.
5. The authors added a derivation showing that the proposed method induces the same rankings as CONTRA.
6. The authors clarified that samples from the prediction region can be easily drawn by sampling first from the latent distribution. They also pointed to experiments demonstrating wall-clock times for sampling.
7. The authors clarified that the baselines use the same training setups.

**Reviewer Scores:**

- AaZM is very likely to increase their score. The authors clarified that finite-sample marginal coverage follows directly from exchangeability of the conformity scores. They also provided evidence of effectiveness across architectures and also in the presence of model misspecification.
- LvS2 is virtually certain to keep their score. Their initial score was already very positive and the authors addressed their concerns about theoretical assumptions and relationship to CONTRA.
- MeP1 is very likely to keep their score. Their review was already positive. The authors adequately addressed their concerns about prediction region computation and training setup.
- EZpP is very likely to increase their score. Their concerns were largely conceptual in nature and were thoroughly addressed by the rebuttal.

---

### Decision · Program_Chairs · 2026-01-26

Accept (Poster)